# TOWARDS GREATER LEVERAGE: SCALING LAWS FOR EFFICIENT MoE LANGUAGE MODELS

**Changxin Tian, Kunlong Chen, Jia Liu, Ziqi Liu, Zhiqiang Zhang**[*]**, Jun Zhou**
Ling Team, Ant Group
{tianchangxin.tcx,lingyao.zzq}@antgroup.com

## ABSTRACT

Mixture-of-Experts (MoE) has become a dominant architecture for scaling Large Language Models (LLMs) efficiently by decoupling total parameters from computational cost. However, this decoupling creates a critical challenge: predicting the model capacity of a given MoE configurations (*e.g.,* expert activation ratio and granularity) remains an unresolved problem. To address this gap, we introduce *Efficiency Leverage (EL)*, a metric quantifying the computational advantage of an MoE model over a dense equivalent. We conduct a large-scale empirical study, training over 300 models up to 28B parameters, to systematically investigate the relationship between MoE architectural configurations and EL. Our findings reveal that EL is primarily driven by the expert activation ratio and the total compute budget, both following predictable power laws, while expert granularity acts as a non-linear modulator with a clear optimal range. We integrate these discoveries into a unified scaling law that accurately predicts the EL of an MoE architecture based on its configuration. To validate our derived scaling laws, we designed and trained `MoE-mini`, a model with only 0.85B active parameters, alongside a 6.1B dense model for comparison. When trained on an identical 1T high-quality token dataset, `MoE-mini` matched the performance of the 6.1B dense model while consuming over 7x fewer computational resources, thereby confirming the accuracy of our scaling laws. This work provides a principled and empirically-grounded foundation for the scaling of efficient MoE models.

## 1 INTRODUCTION

Mixture-of-Experts (MoE) models (Shazeer et al., 2017; Jiang et al., 2024; DeepSeek-AI, 2024) have emerged as a leading paradigm for constructing large language models (LLMs) (Zhao et al.,

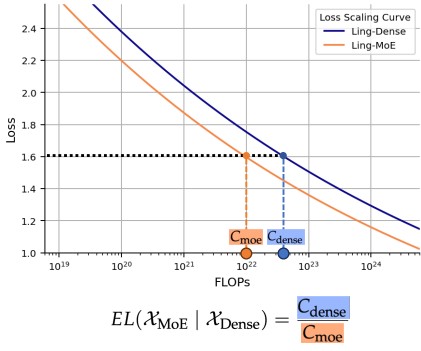

$$EL(\mathcal{X}_{\text{MoE}} \mid \mathcal{X}_{\text{Dense}}) = \frac{C_{\text{dense}}}{C_{\text{moe}}}$$

(a) Definition of Efficiency Leverage (EL)

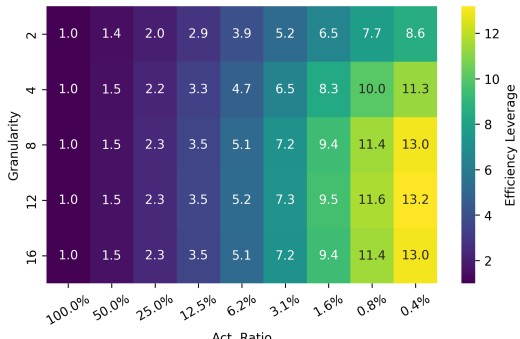

(b) Estimated EL at $1e22$ FLOPs

Figure 1: Illustration of the Efficiency Leverage (EL) metric for MoE architecture versus dense architecture and its estimated values using Eq. 4 for $1e22$ FLOPs.

---

[*]Corresponding author

2023), primarily due to its remarkable computational efficiency (Clark et al., 2022). By leveraging sparse activation, MoE models can dramatically increase their total parameter count without proportionally increasing the computational cost (FLOPs). For instance, DeepSeekMoE (Deepseek-AI et al., 2024), with 16 billion total parameters, activates only 2.8 billion per token, yet achieves performance comparable to a 7-billion-parameter dense model, showcasing a parameter efficiency gain of approximately 2.5x. However, the decoupling of computational cost from the total parameter count in MoE introduces a new challenge in assessing a model's capacity. Specifically, neither the total nor the activated parameter count alone serves as a reliable proxy for performance of MoE models. Consequently, predicting the effective capacity of a specific MoE architecture and setting realistic performance expectations before pre-training remains a critical and unresolved problem. While scaling laws are fundamental for predicting language model performance, their application to MoE models remains fragmented. Prior work has largely studied architectural factors like sparsity or granularity in isolation (Clark et al., 2022; Ludziejewski et al., 2024; Abnar et al., 2025; Ludziejewski et al., 2025). This leaves a critical question unanswered: *how do these factors collectively determine an MoE's true computational advantage over a standard dense model?*

To address this gap, we introduce **Efficiency Leverage (EL)**, a metric that quantifies an MoE's computational advantage over a dense counterpart. As illustrated in Figure 1a, at compute budget $C_{\mathrm{MoE}}$, we define EL as the ratio of training compute budgets a dense model $\mathcal{X}_{\mathrm{Dense}}$ requires to match the performance (*e.g.,* identical loss) of an MoE model $\mathcal{X}_{\mathrm{MoE}}$: $\mathrm{EL}(\mathcal{X}_{\mathrm{MoE}} \mid \mathcal{X}_{\mathrm{Dense}}; C_{\mathrm{MoE}}) = \frac{C_{\mathbf{Dense}}}{C_{\mathbf{MoE}}}$. This definition provides a powerful and intuitive benchmark for MoE architectural comparison: an EL of 5, for example, means an MoE architecture matches the performance of a dense model trained with five times the compute budget. Consequently, for a fixed compute budget, a higher EL directly translates to greater efficiency, enabling larger and more capable models.

To build a predictive framework for EL, our study follows a three-stage methodology. First, we **establish fair training conditions** by deriving scaling laws for hyperparameters and data allocation in preliminary experiments. Second, we **systematically isolate the impact** of core architectural dimensions (such as activation ratio, granularity, and shared experts) on EL. Finally, we **synthesize these findings into a unified scaling law** that accurately predicts an MoE configuration's EL, offering a practical guide for designing next-generation efficient models. Applying this methodology, we trained *over 300 MoE models up to 28B parameters*, using a total of 680k H800-equivalent GPU-hours. This large-scale effort led us to identify several core principles for optimizing the efficiency of MoE models. Our key findings are:

1. **Activation ratio as the primary driver of efficiency.** The expert activation ratio emerges as the primary determinant of EL. We observe a stable power-law relationship: EL increases as the activation ratio decreases (*i.e.,* as sparsity increases). This reveals that sparsely activated pathways yield consistent and predictable gains in computational efficiency.

2. **Expert granularity as a non-linear modulator.** Superimposed on this primary trend, expert granularity introduces a log-polynomial adjustment to EL. This effect is independent of the total compute budget and implies an optimal range for expert size. Our experiments, which utilize a standard load-balancing loss, identify this optimum to be between 8 and 12.

3. **Amplifying effect of the compute budget.** Crucially, the EL of a given MoE architecture is not static; it scales with the training compute budget, also following a power law. This finding underscores the advantage of MoE models in large-scale pre-training, where their efficiency gains become increasingly significant as computational resources expand.

4. **Secondary impact of other architectural factors.** Other design choices, such as shared experts or the specific arrangement of MoE layers, have a secondary impact on EL, as they typically possess broadly applicable, near-optimal settings that require minimal tuning.

Synthesizing these findings, we derive a unified scaling law for EL. This law integrates the effects of compute budget, activation ratio, and expert granularity, providing a predictive framework to guide efficient MoE design. As a practical demonstration, Figure 1b visualizes the predicted EL landscape under a $1e22$ FLOPs budget, highlighting optimal architectural regions.

According to our derived scaling law for EL, we predict that an MoE model with a 3.1 % activation ratio and a granularity of 12 should achieve an efficiency leverage of over 7x at this compute scale. To validate this prediction, we designed and trained `MoE-mini` (17.5B total, 0.85B active

parameters) against a 6.1B dense counterpart on a 1-trillion-token dataset. The results confirmed our hypothesis: `MoE-mini` achieved a lower final training loss and slightly outperformed the dense model across downstream tasks. This outcome empirically validates our law's prediction of a $> 7\times$ efficiency gain. These findings establish our scaling law as a solid theoretical and empirical foundation for designing future large-scale, efficient MoE models.

## 2 PRELIMINARY

### 2.1 MIXTURE-OF-EXPERT TRANSFORMERS.

**Total and Active Parameters.**   We distinguish between a model's *total parameters* ($N$), which include all weights (including all experts), and its *active parameters* ($N_a$), which comprise only the non-expert weights and the subset of experts activated for a given token.

**Routable and Shared Experts.**   An MoE layer contains two types of experts: $E$ *routable experts*, from which a gate dynamically selects $E_a$ per token, and $E_s$ *shared experts*, which are consistently activated for all tokens to process common knowledge.

**Activation Ratio and Sharing Ratio.**   We characterize the expert configuration with two ratios that quantify utilization. The *Activation Ratio (A)*, defined as $A = (E_a + E_s)/(E + E_s)$, measures the overall sparsity of the MoE layer. The *Sharing Ratio (S)*, defined as $S = E_s/(E_a + E_s)$, represents the proportion of activated experts that are shared.

**Granularity of Experts.**   While traditionally the expert dimension ($d_{\text{expert}}$) was tied to the FFN intermediate size (e.g., $4d_{\text{model}}$), recent work decouples them to explore finer-grained experts. We define *Expert Granularity (G)* as $G = 2d_{\text{model}}/d_{\text{expert}}$ to systematically analyze expert size. A higher $G$ indicates a shift towards more, smaller experts for a fixed parameter budget, departing from the conventional practice where $d_{\text{expert}}$ was tied to the FFN's intermediate dimension (e.g., $4d_{\text{model}}$). [1]

**Model Scale in Computation.**   Following prior work (Bi et al., 2024), we define model scale ($M$) as the non-embedding FLOPs per token. This metric provides a fair basis for comparing dense and MoE architectures, as it inherently accounts for sparse activation. The total training compute ($C$) for $D$ tokens is then given by $C = M \cdot D$. We provide the exact calculation for $M$ in Appendix I.

### 2.2 SCALING LAWS FOR MoE OPTIMAL HYPER-PARAMETERS

To ensure fair architectural comparisons, we first establish scaling laws for the optimal training hyperparameters of MoE models. By performing a large-scale hyperparameter search over a wide range of compute budgets ($C$), we derived the scaling laws for the optimal learning rate ($\eta^{\text{opt}}$) and batch size ($B^{\text{opt}}$). Our analysis, detailed in Appendix E.1, reveals a key distinction from dense models: at larger compute scales, MoE models favor a significantly larger batch size and a slightly lower learning rate (Figure 2a). This phenomenon is attributable to MoE's sparse backpropagation, where gradients from only a subset of tokens in a batch update each expert's parameters. We validated that these derived laws are generalizable across MoE models with varying expert activation ratios (shown in Figure 8). This confirms that our findings provide a reliable foundation for exploring diverse MoE architectures under near-optimal training conditions.

### 2.3 SCALING LAWS FOR MoE OPTIMAL MODEL-DATA ALLOCATION

To achieve compute-optimal training, the allocation of a fixed FLOPs budget ($C$) between model size ($M$) and data size ($D$) is critical. We analyze this trade-off for Mixture-of-Experts (MoE) models and compare them against dense models. Our analysis, detailed in Appendix E.2, yields two key insights: First, consistent with prior work (Bi et al., 2024; Hoffmann et al., 2022), the optimal allocation for both MoE and dense models involves splitting the compute budget roughly equally between model and data scaling (i.e., the scaling exponents are close to 0.5). Second, and

---

[1]Our definition ($2d_{\text{model}}/d_{\text{expert}}$) differs from Ludziejewski et al. (2024) ($4d_{\text{model}}/d_{\text{expert}}$) to align with recent models (DeepSeek-AI, 2024; Moonshot-AI, 2025). This choice leads to different observed scaling phenomena.

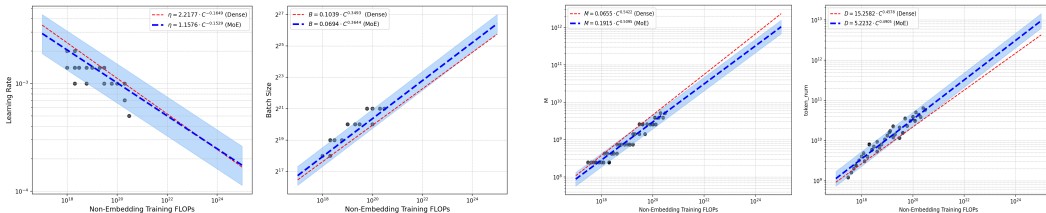

(a) Scaling laws for optimal hyperparameters      (b) Scaling laws for optimal model-data allocation

Figure 2: Scaling laws for optimal hyperparameters and optimal model-data allocation. Blue and red lines represent the fitted laws for MoE and dense models, respectively, derived on the same training dataset. Gray circles are the experimental data points used for fitting.

more crucially, at any given compute budget, the optimal MoE model is computationally smaller but trained on more data than its optimal dense counterpart (Figure 2b). This suggests that MoEs possess greater capacity per parameter, enabling them to effectively leverage larger datasets with smaller model sizes. This finding is particularly significant for training in data-rich but compute-limited scenarios, as it highlights a path toward greater efficiency. *The above scaling laws for optimal hyperparameters and optimal model-data allocation provide a principled basis for model and data selection in our subsequent experiments.*

## 3 EFFICIENCY LEVERAGE: METRIC FOR QUANTIFYING MoE COMPUTE-EFFICIENCY

To quantify the computational advantage of MoEs, we introduce a core metric: **Efficiency Leverage (EL)**. Intuitively, EL measures how many more FLOPs a corresponding dense architecture requires to achieve the same performance as an MoE architecture.

Formally, the EL is defined as the ratio of the compute budgets required for a dense and an MoE architecture to achieve the same target loss value, $L^\star$ at the same dataset. Let $L_\mathcal{X}(C)$ be the optimal loss scaling function for an architecture $\mathcal{X}$, representing the best achievable loss for a given compute budget $C$. Consistent with prior work (Kaplan et al., 2020; Henighan et al., 2020; Achiam et al., 2023), this function is typically modeled as a power law, e.g., $L_\mathcal{X}(C) = \alpha_\mathcal{X} C^{\beta_\mathcal{X}} + b_\mathcal{X}$. The compute required to reach a target loss $L^\star$ is therefore given by the inverse function, $C_\mathcal{X}(L^\star) = L_\mathcal{X}^{-1}(L^\star)$. The EL is then formally expressed as:

$$\text{EL}(\mathcal{X}_{\text{MoE}} \mid \mathcal{X}_{\text{Dense}}; L^\star) = \frac{C_{\mathcal{X}_{\text{Dense}}}(L^\star)}{C_{\mathcal{X}_{\text{MoE}}}(L^\star)}.$$

In our work, we define the target loss $L^\star$ as the loss achieved by the MoE model at its own compute budget, $C_{\text{MoE}}$ (i.e., $L^\star = L_{\mathcal{X}_{\text{MoE}}}(C_{\text{MoE}})$). This practical choice simplifies the EL to a function of the MoE architecture, the dense architecture, and the MoE model's compute budget:

$$\text{EL}(\mathcal{X}_{\text{MoE}} \mid \mathcal{X}_{\text{Dense}}; C_{\text{MoE}}) = \frac{L_{\mathcal{X}_{\text{Dense}}}^{-1}(L_{\mathcal{X}_{\text{MoE}}}(C_{\text{MoE}}))}{C_{\text{MoE}}} = \frac{C_{\text{Dense}}}{C_{\text{MoE}}},$$

where $C_{\text{Dense}}$ is the compute budget required for the dense model to match the MoE's loss, obtained by inverting the dense model's loss scaling curve.

Our primary goal is to build a predictive model for EL based on key MoE architectural choices. We focus on three critical dimensions that govern MoE capacity: the *Activation Ratio* ($A$), *Expert Granularity* ($G$), and *Shared Expert Ratio* ($S$). Other factors, like the arrangement of MoE layers, have a secondary impact (detailed in Appendix F.4). To achieve this, we conduct systematic ablation studies, varying one architectural dimension at a time across a range of compute budgets ($3 \times 10^{18}$ to $3 \times 10^{20}$ FLOPs). Crucially, to ensure a fair comparison, our methodology is guided by the preliminary findings in Sections 2.2 and 2.3. For each experiment, we use our derived scaling laws to set the comparable suboptimal model/data allocation and training hyperparameters. This rigorous protocol ensures that every architecture is evaluated near its peak potential, yielding robust and reliable results. Full experimental details are in Appendix D. The following sections first analyze each factor's impact on EL individually, then synthesize these findings into a unified scaling law.

# 4  SCALING LAWS FOR EFFICIENT MoE ARCHITECTURE

To achieve greater leverage, we first conduct an extensive empirical study on the architectural configurations of MoE and derive unified scaling laws for efficient MoE architectures.

## 4.1  EMPIRICAL STUDY ON THE INTERPLAY BETWEEN LOSS AND MoE ARCHITECTURE

Our investigation focuses on several critical architectural factors: the expert activation ratio ($A$), expert granularity ($G$), and sharing ratio ($S$). For each architectural dimension, we vary it systematically while holding other factors and the model scale $M$ constant. To ensure a fair comparison, all models are trained following the training hyperparameters derived from our scaling laws (Section 2). Guided by the scaling laws for optimal model-data allocation (Section 2.3), we train each model on over three times its optimal number of tokens. This was done to simulate the overtrained state commonly observed in real-world scenarios. A detailed analysis and a complete list of trained models are provided in Appendix F and Appendix J, respectively.

**Expert Activation Ratio ($A$).**  We first investigate the activation ratio ($A$), which governs model sparsity. By varying the total number of experts while keeping the number of activated experts fixed, our IsoFLOPs experiments reveal a clear power-law relationship: for any given computational budget and any given model scale, training loss monotonically decreases with the activation ratio (Figure 3a). This trend holds consistently down to the lowest ratio tested, 1/128 ( 0.8%), demonstrating that greater sparsity yields higher parameter efficiency without an observable turning point. Moreover, this efficiency advantage is amplified at larger training scales, confirming that sparser models are increasingly beneficial in high-computation regimes. See Appendix F.1 for details.

**Expert Granularity ($G$).**  Next, we analyzed expert granularity ($G$), which defines the trade-off between employing numerous small experts versus fewer large ones. Our experiments reveal a distinct U-shaped relationship between granularity and training loss, demonstrating the existence of an optimal point that maximizes performance per FLOP (Figure 3b). This optimum proved to be remarkably stable across different compute budget (*e.g.*, $G$=12 in our tests). This suggests that while overly coarse-grained experts fail to effective specialization (Deepseek-AI et al., 2024), excessively fine-grained experts is also often suboptimal. Crucially, we find that routing quality is a key factor, as poor load balancing shifts the optimal point toward coarser granularities (details in Appendix F.2).

**Shared Expert Ratio ($S$).**  Our analysis of the shared expert ratio ($S$) reveals a U-shaped performance curve, where a small but non-zero ratio minimizes training loss (Figure 3c). Furthermore, we identify a subtle scaling trend: the optimal $S$ decreases as the compute budget grows. This leads to a practical heuristic for large-scale training (*e.g.*, $> 10^{20}$ FLOPs): a "one shared expert" design, representing the minimal effective non-zero ratio, is the most efficient choice (details in Appendix F.3).

**Other Architectural Factors.**  We further analyzed two design dimensions to enhance MoE efficiency: layer arrangement and compute allocation between attention and FFN. We found that incorporating dense layers in the early stages of MoE has minor impact on efficiency but helps mitigate routing imbalances and reduces overall parameters. For compute allocation, allocating 30%-40% of FLOPs to the attention mechanism achieves optimal or near-optimal performance, with minor impact outside this range. Detailed results are available in Appendix F.4.

## 4.2  SCALING LAWS FOR MoE EFFICIENCY LEVERAGE

Based on the empirical study in Section 4.1, shared experts and other design factors have a secondary impact on EL, as they typically have robust, near-optimal settings. Therefore, we focus on deriving a parametric scaling law for EL as a function of activation ratio $A$, granularity $G$, and FLOPs $C$.

### 4.2.1  UNIVARIATE SCALING LAWS FOR EFFICIENCY LEVERAGE

To systematically analyze each core architectural dimension, we vary it while holding the others and the total compute budget (i.e., FLOPs per token, $M$) constant. This controlled approach is essential, as a full combinatorial exploration would be prohibitively complex and unaffordable.

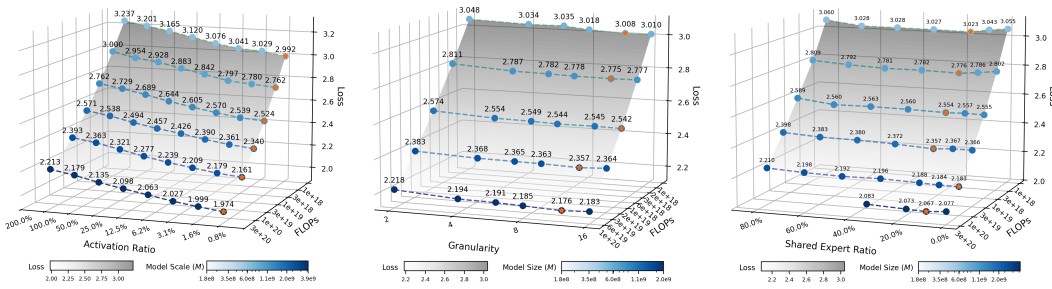

(a) IsoFLOPs curves for varying $A$    (b) IsoFLOPs curves for varying $G$    (c) IsoFLOPs curves for varying $S$

Figure 3: Impact of MoE architectural choices on performance. **(a) Activation Ratio** ($A$)**:** At a fixed compute budget, loss monotonically decreases with a lower activation ratio. The advantage of sparsity is magnified at scale. **(b) Expert Granularity** ($G$)**:** A U-shaped relationship between granularity and loss reveals an optimal point (marked by orange stars) that maximizes efficiency. **(c) Shared Expert Ratio** ($S$)**:** A U-shaped loss curve shows that a low, non-zero $S$ is optimal.

Our procedure for deriving the scaling law for each architectural dimension follows three stages, as detailed in Algorithm 1. First, we generate a dataset of '(compute, loss)' pairs by training a suite of MoE models (Tables 8–10) and their dense counterparts. Second, we fit these points to loss scaling curves for each architectural setting. From these curves, we compute the EL for various MoE architectures and FLOPs budgets, as illustrated in Figures 10b, 11b, and 12b. Finally, we collect the resulting EL values from different settings and use them to derive the univariate scaling laws for activation ratio $A$, granularity $G$, and FLOPs $C$, as presented in Figure 6.

**Interaction of Efficiency Leverage and Activation Ratio.** Our preceding analysis identifies the activation ratio ($A$) as the primary factor influencing EL. As illustrated in Figure 4a, reducing the activation ratio (*i.e.,* increasing sparsity) consistently yields substantial efficiency gains, following a similar power-law relationship across all FLOPs budgets. This leads us to hypothesize: for a given FLOPs budget and granularity, there exists a power-law dependence between EL and activation ratio.

$$\log EL_{C,G}(\hat{A}) = a_A \log \hat{A}, \quad \text{i.e. } EL_{C,G}(\hat{A}) = \hat{A}^{a_A},$$
$$\text{where } \frac{1}{\hat{A}} = \frac{1}{A + (1/A_{start} - 1/A_{max})^{-1}} + \frac{1}{A_{\max}}, \tag{1}$$

where $\hat{A}$ is a saturating transformation of $A$, as defined in Clark et al. (2022), and we set the lower bound of meaningful activation ratio as 0. Clearly, when $A = 1$, we have $EL = 1$, indicating that the EL of the dense model is 1, which satisfies the dense equivalence. We fit Eq. 1 to the data for each compute budget, and the resulting predictions (dotted lines in Figure 4a) align well with our observations. Notably, the fitted exponent $a_A$ is not constant. It increases as $A$ decreases, indicating a diminishing benefit from increased sparsity, consistent with prior work (Clark et al., 2022). Furthermore, $a_A$ also increases with the compute budget $C$, suggesting greater leverage for larger models. We will analyze the relationship between FLOPs and EL in the following paragraph.

**Interaction of Efficiency Leverage and Expert Granularity.** As previously observed, an optimal expert granularity exists that maximizes the EL. Thus, we hypothesize that for a fixed FLOPs budget $C$ and activation ratio $A$, the relationship between EL and $G$ follows a log-polynomial pattern:

$$\log EL_{C,A}(G) = a_G + b_G \left( \log G \left( \log G + c_G \right) \right), \tag{2}$$

where $a_G$ is the granularity-independent base EL, representing the theoretical value when granularity is 1. $b_G$ controls the strength of the curvature in the relationship between EL and granularity, reflecting the sensitivity of the model architecture to changes in expert granularity. $c_G$ determines the position of the optimal granularity that maximizes EL. We fit Eq. 2 to each FLOPs budget and plot the predictions for varying granularity as dotted lines in the Figure 4b. As shown, the curves under different FLOPs budgets are highly similar (*i.e.,* with similar values of $b_G$ and $c_G$), indicating that the impact of expert granularity on MoE efficiency is consistent across various compute budgets.

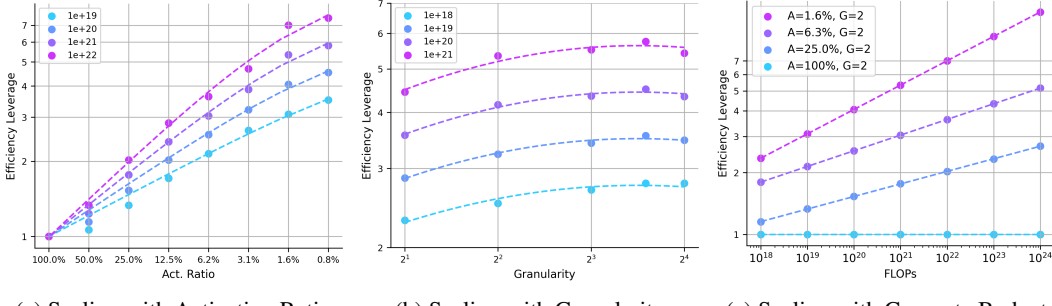

(a) Scaling with Activation Ratio    (b) Scaling with Granularity    (c) Scaling with Compute Budget

Figure 4: Scaling behavior of efficiency leverage (EL). (a) With fixed granularity ($G = 2$), EL follows a power law with respect to activation ratio $A$ across all tested compute budgets ($C$). (b) With a fixed activation ratio ($A = 3.1\%$), EL's scaling with granularity $G$ conforms to a log-polynomial law across all compute budgets. (c) With both activation ratio ($A$) and granularity ($G$) held constant, EL scales with compute according to a standard power law. Points below 3e20 FLOPs represent experimental data, while points beyond this threshold are predictions extrapolated from scaling law.

**Interaction of Efficiency Leverage and Compute Budget.** Based on the analysis presented in Section 4.1 and Section 4.2.1, we observe that the efficiency advantage of MoE increases as the computational budget grows. To formalize the relationship between the FLOPs budget and EL, we assume a standard power-law pattern as follows:

$$\log EL_{A,G}(C) = a_C \log C + c_C, \quad \text{i.e. } EL_{A,G}(C) = \exp(c_C) \cdot C^{a_C}, \tag{3}$$

where $a_C$ reflects the scaling capability of MoE efficiency with respect to the compute budget under given configurations $A$ and $G$. We collect the values of the EL corresponding to different model architectures under the granularity setting of 2, and fit Eq. 2 to each architectures. The predictions for varying granularity are plotted as dotted lines in the Figure 4c. The results indicate that all tested MoE architectures show a trend of higher EL as the FLOPs budget increases, demonstrating the potential of MoE in large-scale pre-training.

Our choice for each univariate scaling law is justified by a goodness-of-fit comparison against simpler alternatives. The specifics of the comparative analysis are presented in Appendix N.

### 4.2.2 JOINT SCALING LAW FOR EFFICIENCY LEVERAGE

Based on the preceding observations and univariate scaling laws, we identify three key insights:

- The activation ratio (or sparsity) is the primary driver of MoE efficiency, establishing a foundational power-law relationship.
- Building upon this power law, expert granularity imposes a non-linear adjustment that operates independently of the compute budget.
- Furthermore, the efficiency advantage of MoE over dense models is amplified by the compute budget $C$ through the power-law pattern.

To unify these interconnected effects, we propose the following joint scaling law for EL:

$$EL(A, G, C) = \hat{A}^{\alpha + \gamma(\log G)^2 + \beta \log G}, \tag{4}$$

where $\alpha = a + d \log C$ is the compute-dependent exponent that captures the primary power-law relationship between EL and FLOPs ratio. The term $a$ represents the base scaling exponent at a reference compute budget, while $d$ is a positive constant that quantifies how the EL is amplified by a larger compute budget $C$. The parameters $\beta$ and $\gamma$ model the non-linear impact of granularity $G$. This quadratic form in $\log G$ directly reflects the log-polynomial pattern observed in our initial analysis, capturing the existence of an optimal granularity.

### 4.2.3 FIT AND VALIDATION

To validate the proposed scaling law for EL, we fit Eq. 4 using Huber loss and the BFGS optimization algorithm (Hoffmann et al., 2022). We use data points with an EL factor below 6 for training,

while those are reserved as a validation set. As depicted in Figure 5, the resulting model achieves an $R^2$ of 0.9858 and demonstrates strong predictive power. This is evidenced by a low RMSE on both the training set (0.2169 over 200 points) and the validation set (0.5275 over 24 points). The quality of the fit is further corroborated by the residuals, which are approximately normally distributed and centered at zero (mean = -0.0273, std. = 0.2803). The fitted coefficients and a more detailed goodness-of-fit analysis can be found in Appendix G and Appendix N.4, respectively.

The alignment between the scaling law and both the training data and validation set provides strong empirical support for the proposed

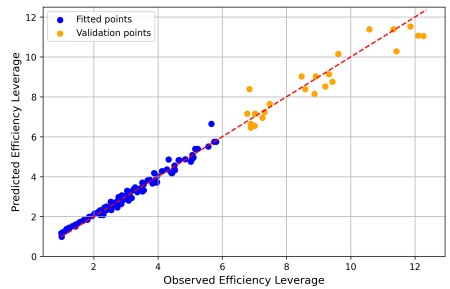

Figure 5: Validation of the Scaling Laws for Efficiency Leverage. We fit Eq. 4 to the data points with an efficiency leverage of less than 6, using the remaining points as the validation set.

relationship. More importantly, the scaling law exhibits remarkable extrapolation capabilities, as it accurately models performance trends for high-leverage validation points outside the training range. These results confirm that Eq. 4 effectively captures the underlying interaction between MoE architecture and EL.

Furthermore, we select $1e22$ FLOPs compute budget, and apply our fitted scaling laws to predict EL across various MoE configurations. As shown in Figure 1, our analysis predicts that an EL exceeding 7x can be achieved at a budget of 1e22 FLOPs with an activation ratio of 3.1% and a granularity of 12. This claim is experimentally validated in the following section.

## 5 MoE-mini: More Efficient MoE Language Model

To validate the scaling laws derived in Section 4, we designed a new MoE model, MoE-mini, configured with architectural parameters predicted to be highly efficient. It features a total of 17.5B parameters but only 0.85B active parameters, achieved through a granularity of $G = 12$ and a low activation ratio of $A = 3.4\%$. Referring to Figure 1, at the $1e22$ FLOPs compute budget, we hypothesize that MoE-mini achieves ***more than 7× in compute-efficiency leverage*** over a comparable dense model. Concurrently, we train a traditional dense model with 6.1 billion parameters (named "Dense-6.1B") for comparison. This section presents a detailed analysis of the performance differences between MoE-mini and the conventional dense model Dense-6.1B, highlighting that the active parameter count, training costs, and downstream inference costs of Dense-6.1B are more than seven times those of MoE-mini. The architectures of MoE-mini and Dense-6.1B are given in Table 1, while ther detailed architectures and training setting are provided in Appendix D.

Table 1: Detailed Architectures of MoE-mini and Dense-6.1B for Comparison.

| Model | $n_{layers}$ | $d_{model}$ | $d_{ffn}$ | $d_{expert}$ | $n_{heads}$ | $n_{kv\_head}$ | $E$ | $E_a$ | $E_s$ | $N$ | $N_a$ |
|---|---|---|---|---|---|---|---|---|---|---|---|
| Dense-6.1B | 28 | 4096 | 14336 | - | 32 | 8 | - | - | - | 6.11B | 6.11B |
| MoE-mini (A0.8B) | 20 | 2048 | 5120 | 384 | 16 | 4 | 384 | 12 | 1 | 17.5B | 0.85B |

### 5.1 Training Dynamics

**The Dynamic of Training Loss** The training loss curves for MoE-mini and Dense-6.1B, shown in Figure 6a, illustrate a clear difference in their convergence behavior. The dense model exhibits faster convergence during the early training phases, indicating an aptitude for rapid initial learning. In contrast, MoE-mini's loss decreases more gradually at the start. However, over the full course of training, MoE-mini steadily improves and ultimately achieves a performance level comparable to that of the dense model, highlighting its ability to reach high performance with sufficient training. Focusing on the final 100 billion tokens of training provides further insight. In this concluding stage, the performance gap between MoE-mini and Dense-6.1B narrows to a negligible difference of about 0.01 in loss value. This confirms that MoE-mini can nearly match the dense

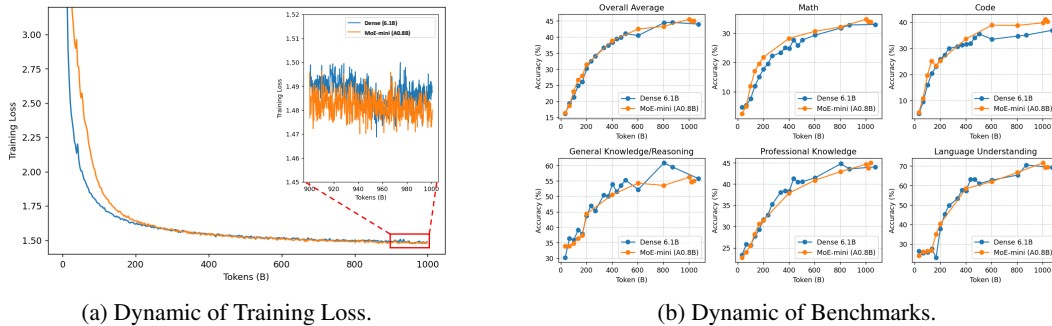



(a) Dynamic of Training Loss.      (b) Dynamic of Benchmarks.

Figure 6: Dynamic of Training Loss (left) and Benchmarks (right).



model's effectiveness while operating with significantly fewer computational resources. Crucially, this near-equal performance underscores `MoE-mini`'s ability to deliver over 7x gains in training efficiency, making it a highly cost-effective and powerful alternative for large-scale pre-training.

**The Dynamic of Benchmarks** Throughout training, `MoE-mini` and `Dense-6.1B` demonstrated remarkably synchronous performance gains on standard benchmarks, shown in Figure 6b. The data reveals a clear and consistent trend: the two models improved almost synchronously. At no point during training did one model show a decisive or lasting advantage over the other. This lockstep progression continued until the end of the training cycle, where they posted nearly identical final scores on the evaluation leaderboard. This synchronous dynamic and convergent outcome suggest a fundamental parity in their learning efficiency and final performance ceiling under our experimental conditions.

## 5.2 EVALUATION

**Evaluation Benchmarks** To provide a holistic assessment of our model's capabilities, we evaluate it on a diverse suite of downstream benchmarks. These tasks are grouped into five key categories: General Knowledge and Reasoning, Language Understanding, Professional Knowledge, Math, and Code. A detailed list of all benchmarks used in each category is provided in Appendix H.

**Evaluation Results** The comparative evaluation, summarized in Table 2, reveals that `MoE-mini` achieves a superior overall score of 45.5, outperforming `Dense-6.1B`'s 44.0. This result demonstrates that `MoE-mini` achieves a "small yet powerful" feat: while its activated parameters constitute only about *13%* of its competitor's during inference, it strikes an exceptional balance between performance and efficiency. Beyond the overall average, `MoE-mini` demonstrates consistent advantages across most key domains, including reasoning, language understanding, code generation, and advanced mathematics. Its superiority is particularly pronounced in tasks requiring high coding proficiency and deep contextual understanding. While there are minor variations on specific benchmarks, the general trend confirms its strong potential in solving complex problems. This result validates that `MoE-mini` achieves an impressive 7× efficiency leverage, delivering performance comparable to a 6.1B dense model that uses over 7 times the active parameters. A detailed, benchmark-by-benchmark comparison is provided in Appendix H.



Table 2: Performance comparison of `MoE-mini` (17B-A0.8B) and `Dense-6.1B`.



| Model | General/Reasoning | Professional | Language | Code | Math | Overall Avg. |
|-------|-------------------|--------------|----------|------|------|--------------|
| **Dense-6.1B** | 55.8 | 44.0 | 69.2 | 36.9 | 32.9 | 44.0 |
| **MoE-mini (A0.8B)** | **56.2** | **44.7** | **71.6** | **39.8** | **34.7** | **45.5** |

## 6 RELATED WORK AND DISCUSSION

We provide a broader survey of related work in Appendix B and compare our findings with key prior studies in Appendix C. Our work formulates scaling laws in terms of Efficiency Leverage (EL), diverging from prior loss-centric studies. This EL-based approach offers a more direct and practical framework for understanding MoE efficiency for two key reasons: **1) EL directly quantifies an MoE's compute advantage.** Unlike absolute loss, which is dataset-specific and hard to interpret, EL provides a generalizable architectural insight. **2) EL can Simplify Model Selection.** Instead of fitting multiple complex loss functions, practitioners can use our scaling laws to directly compare the efficiency of different MoE configurations. This dramatically simplifies architectural design choices. In short, while traditional laws predict *what* the loss will be, our formulation quantifies *how much more efficient* an MoE architecture is, offering actionable design guidance.

## 7 LIMITATIONS AND FUTURE WORK

Our study has four primary limitations, which also point to valuable directions for future research.

First, following standard practice (Clark et al., 2022; Kaplan et al., 2020; Hoffmann et al., 2022), we measure computational cost in theoretical FLOPs. This hardware-agnostic metric overlooks practical wall-clock effects (communication, memory, kernel efficiency, parallelization). Our work thus establishes a theoretical upper bound on efficiency, a necessary first step before optimizing for real-world costs. Second, to make a systematic analysis feasible, we assume that MoE architectural factors are independent. This allowed us to pragmatically study each factor in isolation and synthesize the results into a unified law. However, this approach may overlook interaction effects that could unlock further optimizations. Third, due to resource constraints, we applied a single hyperparameter scaling law to all MoE models, regardless of their sparsity. While effective, developing a sparsity-aware hyperparameter law that tailors settings to each model is a promising avenue for future work. Fourth, our scaling laws focus on compute budget rather than its allocation between training data and model size. Establishing a *Chinchilla-like* scaling law in term of model size and dataset size for MoEs' efficiency to guide this trade-off is an important next step.

Despite these limitations, our findings confirm the significant potential of MoE models, which provide a clear path toward more capable and efficient models in terms of theoretical compute cost.

## 8 CONCLUSION

In this work, we introduce Efficiency Leverage (EL), a metric quantifying an MoE model's computational advantage over a dense counterpart, to analyze how architectural choices govern performance. Our large-scale study of over 300 models reveals that MoE efficiency follows predictable principles: EL scales as a power-law with activation ratio and compute budget, while expert granularity has a non-linear effect with a distinct optimal range. Other factors, like shared experts, have a secondary impact. We unified these principles into a single scaling law that accurately predicts MoE efficiency. To validate it, we designed a 0.85B activated parameter MoE model which, as predicted, achieved over 7x efficiency leverage, confirming our law's robust predictive power. For future work, our framework can be extended in several key directions: (1) Incorporating memory constraints and communication overhead into the EL framework, particularly for distributed training scenarios where these factors dominate practical efficiency. (2) Developing a unified metric that balances training compute with inference latency, enabling end-to-end efficient architecture co-design. We hope this work inspires continued innovation in MoE architectures toward greater leverage.

## USE OF LARGE LANGUAGE MODELS

During the preparation of this work, we used LLMs (*e.g.,* GPT-5 and Gemini-2.5-pro) to assist with editing and polishing the manuscript for clarity and readability. Furthermore, the plotting code for the figures presented in this paper was generated with the assistance of these models.

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

## A    NOTATION

To aid readability, we provide a list of key symbols used throughout this paper.

Table 3: Notation.

| Symbol | Description |
|--------|-------------|
| $E$ | Number of routable experts. |
| $E_a$ | Number of activated experts. |
| $E_s$ | Number of shared experts. |
| $N$ | Number of non-vocabulary parameters. |
| $N_a$ | Number of activated parameters. |
| $d_{model}$ | Model hidden dimension. |
| $d_{expert}$ | Expert hidden dimension. |
| $C$ | Total training compute in FLOPs |
| $M$ | Compute (w/o embedding) per token in FLOPs. |
| $D$ | Dataset size in tokens. |
| $A$ | Activation ratio, *i.e.,* $(E_a + E_s)/(E + E_s)$. |
| $G$ | Granularity of experts, *i.e.,* $2d_{mode;}/d_{expert}$ |
| $S$ | Shared expert ratio, *i.e.,* $E_s/(E_a + E_s)$ |

## B    RELATED WORK

### B.1    SCALING LAWS FOR LANGUAGE MODELS

Scaling laws provide a framework for understanding and predicting the performance of language models under varying conditions. Kaplan et al. (2020) laid the foundation by demonstrating that model performance adheres to predictable power-law relationships involving model size, dataset size, and compute budget. Building on this, Hoffmann et al. (2022) introduced the Chinchilla scaling laws, highlighting the importance of balancing model size and training data volume for compute-optimal training. They showed that scaling model size without a corresponding increase in data leads to diminishing performance gains. Sardana et al. (2023) advanced this understanding by incorporating inference costs into compute-optimal frameworks, proposing strategies for optimizing performance under fixed inference constraints. Additionally, Bi et al. (2024) emphasized the critical role of data quality, demonstrating that higher-quality datasets enable more efficient scaling, particularly with larger models. Recent advancements have applied these scaling laws to various specialized areas. For example, hyperparameter optimization has been explored in the context of scaling laws (Bi et al., 2024; Li et al., 2025), while Gadre et al. (2024) investigated the phenomena of over-training and its implications on model performance. Furthermore, scaling laws have been analyzed for their impact on downstream task performance across a range of applications (Chen et al., 2024; Ruan et al., 2024; Isik et al., 2025; Hu et al., 2023; Grattafiori et al., 2024; Li et al., 2025), underscoring their adaptability and relevance in addressing both theoretical and practical challenges in language modeling.

### B.2    SCALING LAWS FOR MIXTURE-OF-EXPERTS (MOE)

Mixture-of-Experts (MoE) models (Shazeer et al., 2017; Lepikhin et al., 2020) have emerged as a powerful architecture for language modeling, primarily due to their ability to decouple computational cost from parameter count. Recent research has further explored optimizations within the MoE

paradigm. For instance, DeepSeekMoE (Deepseek-AI et al., 2024) investigated the impact of fine-grained expert settings on model performance, proposing a novel design that incorporates shared experts and a hybrid structure combining dense layers with MoE layers. Complementing this, Zoph et al. (2022) highlighted that the performance gains from increased sparsity diminish significantly once the number of experts exceeds 256, suggesting a practical limit for highly sparse models. With the widespread adoption of the MoE architecture, the scaling laws governing MoE models have been extensively studied. Early work by Clark et al. (2022) examined scaling by varying model size and the number of experts on a fixed dataset, concluding that routed models offer efficiency advantages only up to a certain scale. This analysis was subsequently extended by Ludziejewski et al. (2024), who incorporated variable dataset sizes and explored the effects of expert granularity. Additionally, Wang et al. (2024a) investigated the transferability and discrepancies of scaling laws between dense models and MoE models. Abnar et al. (2025) advanced this line of inquiry by deriving scaling laws for optimal sparsity, explicitly considering the interplay between training FLOPs and model size. They also analyzed the relationship between pretraining loss and downstream task performance, noting distinct behaviors between MoE and dense models on certain tasks. More recently, Ludziejewski et al. (2025) derived joint scaling laws applicable to both dense Transformers and MoE models, demonstrating that MoE architectures can outperform dense counterparts even under constraints of memory usage or total parameter count. Liew et al. (2025) derive empirical scaling laws for upcycling LLMs to MoE models, relating performance to both dataset size and architectural choices.

## C  COMPARISON WITH PREVIOUS WORKS.

**Comparison with Clark et al. (2022).**  Clark et al. (2022) used a fixed dataset and concluded that the efficiency of MoE models over dense models diminishes beyond a certain scale. In contrast, our results (Figure 10) demonstrate that MoE models are consistently more compute-efficient across all scales we tested. The discrepancy may lie in their experimental design: using a fixed dataset. As our scaling laws establish (Section 2.3), MoE models require proportionally more training data than dense models for compute-optimal training. A fixed dataset therefore systematically under-trains MoEs, leading to an unfair comparison and flawed conclusions. Our convergence curves (Figure 6a) and findings from Ludziejewski et al. (2024) confirm this: MoEs, despite a slower start, eventually surpass dense models. Unlike prior work, we follow scaling laws to allocate resources, dynamically scaling training tokens with compute. This ensures the fairness and reliability of our comparison.

**Comparison with Ludziejewski et al. (2024).**  Our findings on expert granularity differ from Ludziejewski et al. (2024) in two key ways. First, we find a log-polynomial relationship suggesting an optimal granularity, not their reported monotonic trend where finer is always better. Second, our MoE's efficiency loss (EL) is typically under 10x, substantially lower than their reported ¿10x "Relative FLOPs to train equivalent Transformer". These discrepancies stem from three core differences in experimental design: (1) Granularity definition: Our definition ($G = 2d_{\text{model}}/d_{\text{expert}}$), aligned with leading models (DeepSeek-AI, 2024; Moonshot-AI, 2025), uses experts half the size of theirs at the same nominal granularity. This allows us to test a truly finer spectrum. (2) Hyperparameter strategies: We optimize hyperparameters for each compute budget, unlike their fixed-setting approach, which is crucial for fair comparison as optimal settings vary with scale (Section 2.2). (3) Base MoE architectures: Our MoE uses a denser activation ratio (1/32 vs. their sparser 1/64). Their inherently more efficient baseline may inflate their reported gains. In summary, our differing conclusions arise from exploring a finer granularity spectrum under fairer, optimized training conditions.

**Comparison with Abnar et al. (2025).**  While our findings align with Abnar et al. (2025) on the principle that larger, sparser models perform better under a fixed compute budget, our work extends theirs in two crucial ways. First, methodologically, we optimize training hyperparameters and systematically analyze architectural factors like expert granularity, uncovering its log-polynomial effect on performance. Second, and more importantly, our primary contribution is the derivation of a novel scaling law for the *efficiency leverage* of MoE models over their dense counterparts, rather than for loss. This law's key advantage is its independence from specific datasets. It directly quantifies the relationship between MoE architecture and relative efficiency, yielding more generalizable and actionable principles for model design.

**Comparison with Ludziejewski et al. (2025).** Our work and Ludziejewski et al. (2025) are complementary, as we investigate different aspects of MoE scaling laws. We focus on optimizing architectural parameters (*i.e.,* granularity, activation ratio) within a fixed compute budget and model scale. They, in contrast, determine the optimal allocation between model size and data volume under both compute and memory constraints. While we also explored model-data allocation, our analysis was intentionally limited. Its purpose was not to derive a comprehensive allocation strategy, but rather to establish that MoE and dense models have fundamentally different resource needs. This foundational insight was critical, justifying our approach of providing ample, near-optimal training budgets to ensure a fair and reliable comparison across all models in our main experiments.

**Reconciling Findings on Shared Expert Effectiveness with OLMoE(Muennighoff et al., 2024).** Contrary to the findings of Muennighoff et al. (2024) with OLMoE, our scaling law analysis suggests a shared expert is generally beneficial. We attribute this discrepancy primarily to our broader scope of analysis and distinct model architecture. While the OLMoE conclusion stems from a single data point, ours is derived from a trend across numerous models and scales. This broader perspective reveals that although specific configurations in our study perform best without a shared expert (e.g., M=2e9, Figure 3c)—aligning with OLMoE's observation—the dominant trend favors its use. Furthermore, our 256-expert architecture features significantly higher sparsity and finer granularity than OLMoE's, a key structural difference that, along with varying training parameters, can alter its impact. Therefore, we conclude that while its benefit is context-dependent, a shared expert is a robust choice from a general scaling perspective.

## D EXPERIMENTAL SETUP

**Architecture and Tokenizer** We adopt a Grouped Query Attention (GQA) (Ainslie et al., 2023) architecture based on the standard decoder-only Transformer, consisting of an embedding layer, multiple alternating layers of attention mechanisms and feed-forward networks, and a final de-embedding layer. Additionally, we use the BPE (Byte-Pair Encoding) algorithm (Sennrich et al., 2015) and RoPE (Rotary Positional Embedding) (Su et al., 2024) to handle positional information. The vocabulary size is 126,464, and the sequence length is 4,096.

**Expert Routing Strategy** In our MoE layers, a routing network assigns each token's hidden state $h_t$ to the top-$N_a$ experts. This is achieved by generating gating scores $g_t = \text{Softmax}(W_g \cdot h_t)$, where $W_g$ is a learnable matrix. The final output is a weighted sum of the selected experts' outputs: $o_t = \sum_{i \in \text{TopK}(g_t)} g_{t,i} \cdot E_i(h_t)$, where $E_i$ is the $i$-th expert in total $N$ experts. To ensure balanced expert utilization and stable training, we incorporate two standard auxiliary losses: a load balancing loss (Lepikhin et al., 2020) (coefficient of 0.01) to encourage uniform token distribution, and a router z-loss (Zoph et al., 2022) (coefficient of 0.001) to regularize the magnitude of the gating logits.

**Optimizer and Scheduler** The parameters of experimental models are initialized from a distribution with a standard deviation of 0.006 and optimized using the AdamW optimizer (Loshchilov & Hutter, 2017). The optimizer's hyperparameters are set to $\beta_1 = 0.9$ and $\beta_2 = 0.95$, with 0.1 weight decay applied. The learning rate schedule employs a WSD (warmup-stable-decay) strategy (Hu et al., 2024): the first 1% of training steps use linear warm-up, followed by exponential decay that reduces the learning rate to 10% of its peak value.

**Pre-training Data** The training data is sourced from a large-scale multilingual corpus, primarily covering English and Chinese, while also including various other languages. This corpus encompasses web text, mathematical materials, programming scripts, published literature, and diverse textual content. To validate model performance, we extracted a 2T-token subset from this corpus for training. In Table 4, we present the composition of the training datasets for all experiments. Unless otherwise specified, this configuration is used throughout.

Table 4: Pre-training data composition.

| Type | Web | Books | Wiki | Academic | Code | News | Social | Domain | SFT | Math | Exam |
|------|-----|-------|------|----------|------|------|--------|--------|-----|------|------|
| **Ratio** | 46.0% | 5.0% | 4.0% | 6.0% | 25.0% | 0.1% | 1.9% | 1.0% | 4.0% | 6.0% | 1.0% |

**Other Training Configurations**   Our implementation is built on Megatron-LM and employs a hybrid parallel strategy combining Expert Parallelism (EP), Tensor Parallelism (TP), and Pipeline Parallelism (PP). We utilized bfloat16 precision for all forward and backward passes to maximize throughput, while maintaining float32 for the master weights and optimizer states to ensure numerical stability. The representative parallelism configurations in our experiments are as follows.

# E    DETAILED PRELIMINARY EXPERIMENTS

## E.1    SCALING LAWS FOR MoE OPTIMAL HYPER-PARAMETERS

The performance of a MoE model is sensitive to its hyperparameters. To ensure that our subsequent architectural comparisons are reliable, it is crucial to evaluate each configuration under its optimal hyperparameter settings. Therefore, we first conduct a preliminary study to establish the scaling laws for optimal MoE hyperparameters. Previous research (Bi et al., 2024) has established that the optimal hyperparameters are primarily a function of the total computational budget. Accordingly, we performed a hyperparameter search across a compute range of $3e17$ to $3e20$ FLOPs, using a Warmup-Stable-Decay (WSD) learning rate schedule (Hu et al., 2024). We trained multiple models, varying both learning rate and batch size, which were sampled from a log-base-2 grid. Specifically, the exponents for the learning rate ranged from -11 to -9.0, and for the batch size, from 18 to 21. To make this analysis tractable, we initially fixed the MoE configuration to one with 64 experts, of which 4 are activated per token, plus an additional shared expert (resulting in an activation ratio $A = 7.8\%$ and a granularity $G = 2$). Detailed settings of the experimental models are available in the Appendix D. We then verified that the conclusions from this configuration generalize across different activation ratios.

Figure 7 illustrates the fitting process. To ensure robustness, we identify "near-optimal" configurations as those achieving a loss within 0.25% of the minimum for a given compute budget. After removing outliers, we fitted the optimal batch size, $B^{\mathrm{opt}}$, and learning rate, $\eta^{\mathrm{opt}}$, against the compute budget $C$. The resulting scaling laws reveal clear trends: $B^{\mathrm{opt}}$ increases and $\eta^{\mathrm{opt}}$ decreases with larger $C$. The final formulas obtained from the fitting process are as follows:

$$\eta^{\mathrm{opt}} = 1.1576 \cdot C^{-0.1529}$$
$$B^{\mathrm{opt}} = 0.0694 \cdot C^{0.3644}$$

$$(5)$$

A key finding emerges when comparing these laws to those of dense models. As shown in Figure 7, MoE models favor a significantly larger batch size and a slightly lower learning rate at large compute scales. This phenomenon is attributable to MoE's sparsity: during backpropagation, each expert's parameters are updated using only a subset of the tokens in a batch, whereas dense parameters receive gradients from the entire batch (Sun et al., 2024).

To validate the generalizability of these laws, we conduct experiments on MoE models with varying activation ratios. We used the derived laws to predict optimal hyperparameters at a compute budget of $3e20$ FLOPs, after fitting them on data up to $1e20$ FLOPs. As shown in Figure 8, the predicted optimal regions effectively capture the best-performing hyperparameters for activation ratios from 4.7% to 10.9%, demonstrating that the laws can be applied to MoE models within this range of activation rates. This confirms that our hyperparameter scaling laws provide a reliable foundation for exploring diverse MoE architectures under fair and near-optimal training conditions.

## E.2    SCALING LAWS FOR MoE OPTIMAL MODEL-DATA ALLOCATION

To determine optimal allocation between model size and data size, we analyze loss trajectories across FLOPs budgets from hyperparameter scaling experiments. By identifying the $(M, D)$ combination that yields the minimum loss for a fixed FLOP budget, we derive optimal allocation strategies for

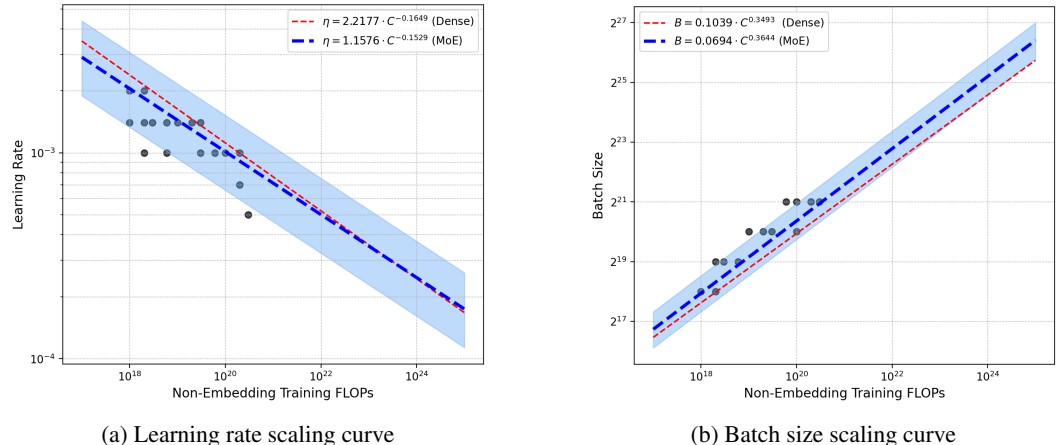

(a) Learning rate scaling curve

(b) Batch size scaling curve

Figure 7: Scaling laws for optimal hyperparameters. Blue and red lines represent the fitted laws for MoE and dense models, respectively, derived on the same training dataset. Gray circles are the experimental data points used for fitting.

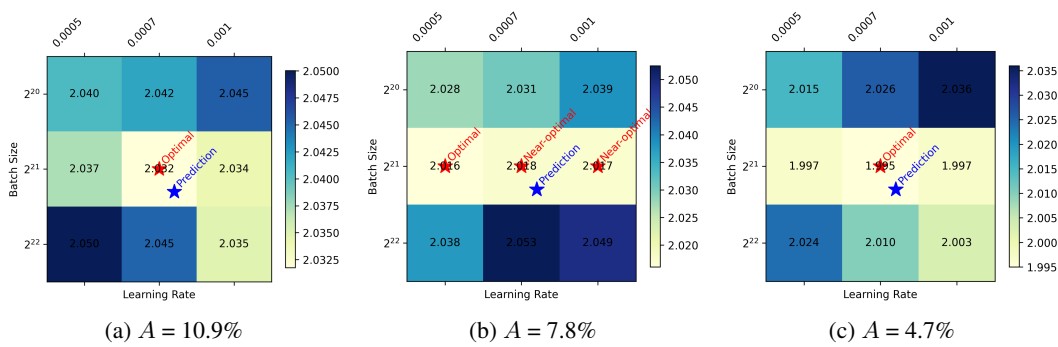

(a) $A = 10.9\%$

(b) $A = 7.8\%$

(c) $A = 4.7\%$

Figure 8: Validation of MoE hyperparameters scaling laws across different activation ratios ($A$). "*Near-optimal*" refers to hyperparameters achieving a loss within 0.25% of the optimal ones.

specific MoE configurations activating 4 of 64 experts and an additional shared expert ($A = 7.8\%$, $G = 2$). Crucially, MoE capacity exhibits strong dependence on activation ratio. Thus, this analysis aims to deepen our understanding of MoE architectures and to provide general guidance for model selection in subsequent experiments. The problem can be formally defined as:

$$(M^{\text{opt}}, D^{\text{opt}}) = \arg \min_{M,D} \mathcal{L}(M, D; C, A, G, S) \quad \text{s.t.} \quad C = M \cdot D \tag{6}$$

The resulting scaling laws for the optimal model size ($M^{\text{opt}}$) and data size ($D^{\text{opt}}$) are presented in Figure 9 and summarized in Table 5. For comparison, we derive the same laws for dense models. Our analysis yields two key insights:

1. The optimal allocation coefficients for different architectures are similar and close to 0.5. This aligns with findings from previous studies (Bi et al., 2024; Hoffmann et al., 2022), indicating that for compute-optimal training, the budget should be split roughly equally between increasing model size and data volume.

2. Crucially, at any given compute budget, the optimal MoE model is computationally smaller (lower $M^{\text{opt}}$) but trained on more data (larger $D^{\text{opt}}$) than its optimal dense counterpart. This suggests that MoEs possess greater capacity, enabling them to support larger training datasets with smaller model sizes. In real-world scenarios where data is abundant but computational resources are limited, this is significant for improving efficiency.

While practical training strategies may deviate from this compute-optimal allocation, these scaling laws provide a crucial reference. They offer a principled basis for determining the necessary amount of training data for a given model to approach convergence, designing informative ablation studies, and ultimately, developing more efficient MoE architectures.

Table 5: Scaling law parameters for compute-optimal allocation of model scale ($M^{\text{opt}}$) and data size ($D^{\text{opt}}$) for MoE and dense models on identical datasets.

|  | Optimal Model Scale ($M^{\text{opt}}$) | Optimal Data Size ($D^{\text{opt}}$) |
|---|---|---|
| Dense | $M^{\text{opt}} = 0.0655 \cdot C^{0.5422}$ | $D^{\text{opt}} = 15.2582 \cdot C^{0.4578}$ |
| MoE | $M^{\text{opt}} = 0.1915 \cdot C^{0.5095}$ | $D^{\text{opt}} = 5.2232 \cdot C^{0.4905}$ |

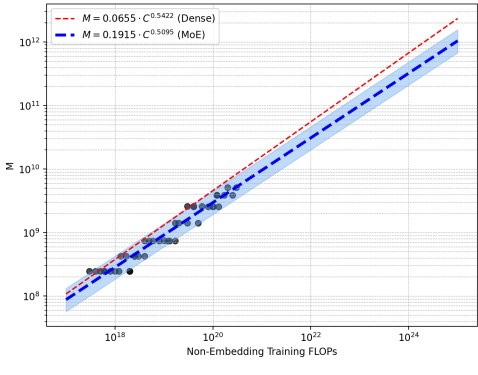 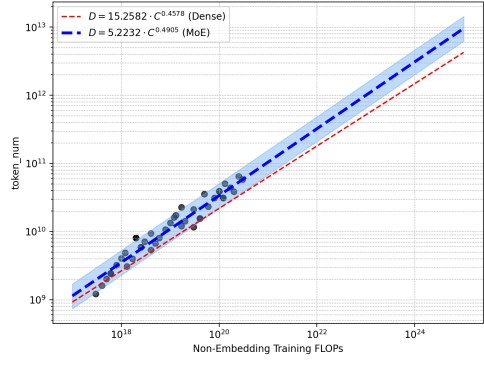

(a) Optimal Model Scale ($M^{\text{opt}}$) Scaling      (b) Optimal Data Size ($D^{\text{opt}}$) Scaling

Figure 9: Scaling laws for optimal model scale ($M^{\text{opt}}$) and data size ($D^{\text{opt}}$) on identical datasets. For a given budget, MoE models (blue) optimally allocate more resources to data and fewer to model size compared to dense models (red).

## F   DETAILED EXPERIMENTAL ANALYSIS OF MoE ARCHITECTURE

### F.1   OPTIMAL EXPERT ACTIVATION RATIO

We begin by investigating the activation ratio ($A$), a critical factor governing MoE efficiency. Our experimental design isolates the effect of $A$ by holding the computational cost per token ($M$) con-

stant. This is achieved by fixing the number of activated experts and their granularity, while varying the total number of experts in the pool from 2 to 256. This setup allows us to explore a wide range of activation ratios (from 0.8% to 100%, where 100% represents a dense model) without altering the forward pass FLOPs. The optimization problem for a given compute budget $C$ is thus:

$$A^{\text{opt}} = \arg\min_A \mathcal{L}(A; C, M, G, S) \tag{7}$$

The IsoFLOPs curves, presented in Figure 10a, reveal a clear and consistent trend. Across all tested FLOPs budgets (from $1e18$ to $3e20$), loss monotonically decreases with activation ratio, following a power-law pattern. For all configurations, the lowest tested ratio of 0.8% consistently yields the minimum loss. This finding suggests a core principle: for a fixed computational cost, greater model sparsity (*i.e.,* lower activation ratio) leads to higher parameter efficiency.

To quantify this efficiency improvement, we fit a series of loss scaling curves at different activation ratios. Based on these curves, we compute the efficiency leverage for different activation ratios and FLOPs budgets, as illustrated in Figure 10b. The results reveal two key trends. First, for a fixed FLOPs budget, the EL consistently increases as the activation ratio decreases, indicating that sparse activation can always enhance computational efficiency. Second, for a fixed activation ratio, the EL grows with the computational budget, demonstrating that the MoE advantage is amplified at larger scales. These findings confirm that reducing the activation ratio yields substantial efficiency gains, and these benefits are magnified in large-scale, high-computation regimes.

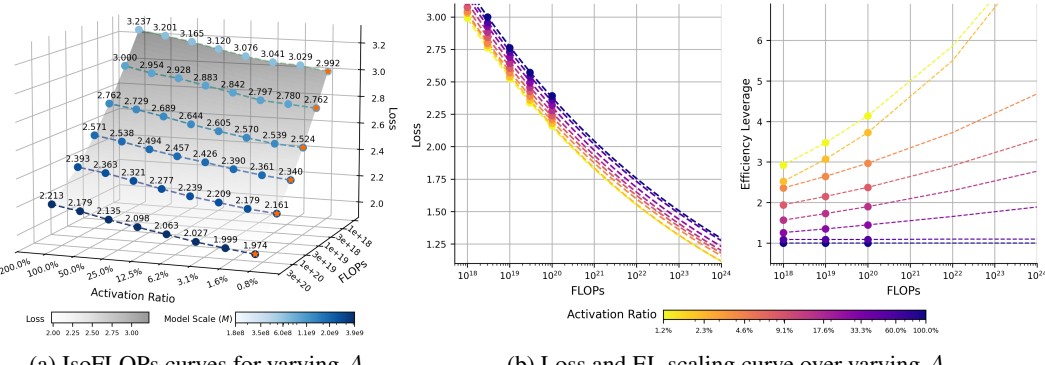

(a) IsoFLOPs curves for varying $A$      (b) Loss and EL scaling curve over varying $A$.

Figure 10: Impact of the Activation Ratio $A$ on Loss and Efficiency. (a) At any fixed compute budget (each colored line), lower activation ratios yield lower loss. The orange stars mark the optimal loss point. (b) Loss and EL scaling curves illustrate that EL increases with both higher compute budgets and lower activation ratios, showing that MoE advantages are magnified at scale.

---

**Key Takeaway 1**

- **Monotonic Relationship Between Efficiency and Activation Ratio.** For a fixed computational cost, model performance consistently improves as the activation ratio decreases. This indicates a direct, monotonic relationship between sparsity and efficiency.

- **Efficiency Gains Amplify with Scale.** The efficiency advantage of MoE models (their EL) grows with the total training budget. This highlights their suitability for large-scale training, where their benefits become even more significant.

---

## F.2 OPTIMAL GRANULARITY OF EXPERTS

The granularity of experts is a critical factor in the efficiency of MoE. While prior works (Ludziejewski et al., 2024; Deepseek-AI et al., 2024) suggests that finer-grained experts improve performance, the optimal balance remains an open question. To investigate the influence of expert granularity on MoE efficiency, for a fixed model size $M$ and activation ratio $A$, we vary the expert granularity from 2 to 16 by increasing the total number of experts from 64 to 512 while proportionally decreasing the

size of each expert to keep computational cost (FLOPs) per token constant. This creates a spectrum of models from coarse-grained (fewer, larger experts) to fine-grained (more, smaller experts). By training these models and comparing their final training losses, we can identify the granularity that yields the best performance for a given FLOPs budget. This problem is formalized as:

$$G^{\text{opt}} = \arg\min_G \mathcal{L}(G; C, M, A, S) \tag{8}$$

where $G^{\text{opt}}$ is the optimal granularity that minimizes the training loss $\mathcal{L}$ under a fixed FLOPs budget $C$, model size $M$, activation ratio $A$, and shared expert ratio $S$. As shown in Figure 11a, our experiments across a range of FLOPs budgets ($10^{18}$ to $10^{20}$) reveal a distinct trend. For any given budget, as we increase expert granularity, the training loss first decreases and then, after reaching a minimum, begins to increase. This demonstrates the existence of an optimal expert granularity that maximizes computational efficiency of MoE. To further analyze this relationship, we fit loss scaling curves for different granularities (Figure 11b), quantifying their impact on EL.

Our study yields two primary insights: First, for a fixed FLOPSs budget, the training loss follows a U-shaped (polynomial) relationship with respect to expert granularity, which confirms an optimal point for maximizing model performance per FLOP. This finding contrasts with the conclusions of Ludziejewski et al. (2024), and we detail the reasons for this discrepancy in Section C. Second, across different FLOPSs budget, the optimal granularity remains within a stable range (around 12 in our experiments), offering a reliable heuristic for model design. Furthermore, we find that routing balance significantly impacts the choice of optimal granularity. Poor routing balance shifts the optimal point towards coarser granularities and degrades overall model performance (see Appendix F.4 for details). This suggests that improving routing mechanisms could unlock the potential of even more fine-grained MoEs, marking a promising direction for future work.

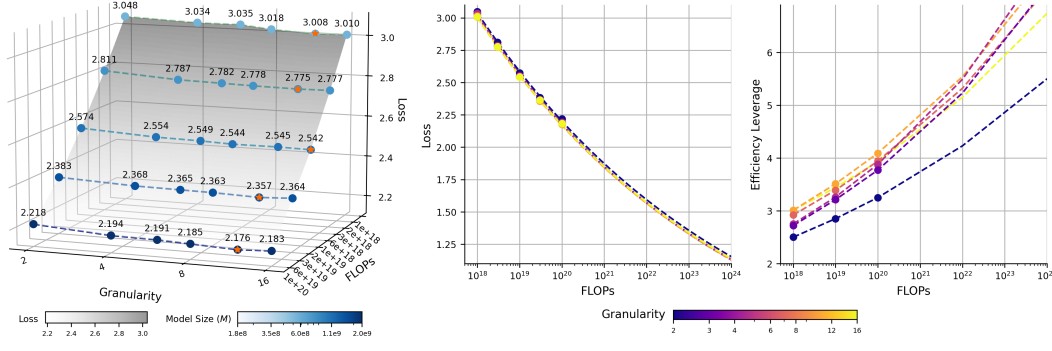

(a) IsoFLOPs curves over varying $G$.     (b) Loss and efficiency leverage scaling curve over varying $G$.

Figure 11: Impact of the Experts Granularity $G$ on Loss and Efficiency. (a) IsoFLOPs curves reveal a U-shaped (polynomial) relationship between expert granularity and training loss. Orange stars mark the optimal granularity for each FLOPs budget. (b) Loss and EL scaling curves show that MoE efficiency improves as FLOPs increase and expert granularity approaches the optimal range.

---

**Key Takeaway 2**

- **Existence of Optimal Expert Granularity.** For a fixed FLOPs budget and model scale, training loss exhibits a U-shaped (polynomial) relationship with expert granularity, indicating an optimum that maximizes efficiency.

- **Stable Range of Optimal Expert Granularity.** The optimal granularity (*e.g.,* around 12 in our experiments) is stable across a wide range of FLOPs budgets. However, poor routing balance shifts this optimum toward coarser granularity.

---

**The Impact of Routing Balance on the Optimal Expert Granularity.** To investigate how routing quality influences the optimal expert granularity, we induce a state of routing imbalance. This is achieved by setting the coefficient of load balancing loss to 0.001, a setup known to cause load imbalance. In this setting, we train MoE models with a varying expert granularity while maintaining a

constant total parameter count. As shown in Figure 12, our results reveal that a coarser expert granularity becomes optimal under such imbalanced routing. Specifically, the IsoFLOPs curves (Figure 12a) demonstrate that models with coarser granularity ($G = 6, 8$) achieve lower loss for a given computational budget. This trend is consistently observed in the loss scaling curves (Figure 12b). This phenomenon indicates that when the routing mechanism becomes a performance bottleneck, a fine-grained architecture with numerous specialized experts is counterproductive. The weakened router cannot distribute tokens effectively, nullifying the benefits of specialization. Consequently, the model benefits more from a coarser-grained design with fewer, more generalized experts, as this simplifies the routing task and mitigates the detrimental effects of the load imbalance.

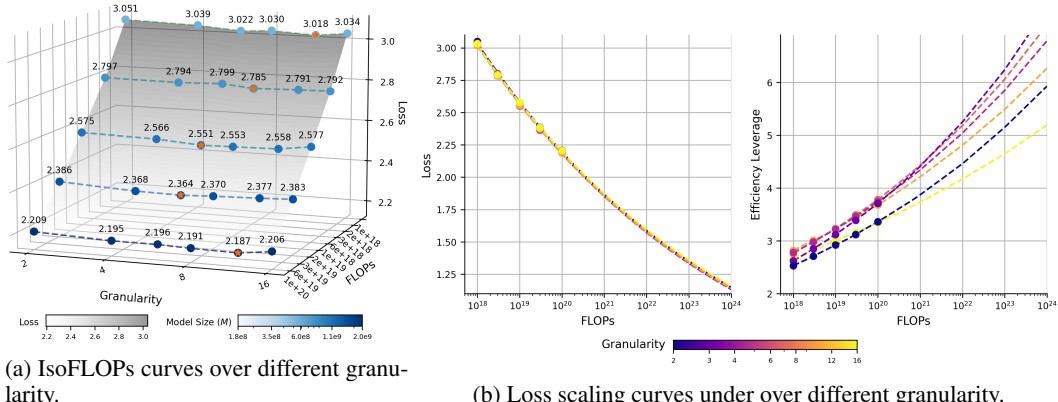

(a) IsoFLOPs curves over different granularity.

(b) Loss scaling curves under over different granularity.

Figure 12: Impact of Expert Granularity on Loss Under Weakened Routing Balance.

**Summary of Expert Granularity Analysis.** Our analysis of expert granularity reveals a trade-off between specialization and load balancing. On one hand, prior work has shown that finer-grained partitioning enhances expert specialization (Deepseek-AI et al., 2024). On the other hand, our experiments demonstrate that increasing granularity leads to a greater load imbalance. We quantify this imbalance using the coefficient of variation (CV) of the expert loads, where a higher value indicates a greater imbalance. As detailed in Table 15, increasing the number of experts ($E$) directly correlates with a higher CV. These results were obtained while holding the balancing loss coefficient constant, in order to isolate the effect of granularity. This controlled setting highlights the inherent challenge of routing tokens to a larger set of smaller experts, thereby distinguishing the benefits of specialization from the practical difficulties of utilization.

### F.3 OPTIMAL SHARED EXPERT RATIO

Shared experts are always active to capture common knowledge (Deepseek-AI et al., 2024). To determine the optimal proportion of shared experts, we designed a series of experiment to isolate the impact of the shared expert ratio $S$. We fix the total model size $M$, the activation ratio $A$, and the total number of active experts ($E_s + E_a$). We then systematically vary $S$ by substituting routed experts ($E_a$) with shared experts ($E_s$), exploring configurations from fully specialized ($S = 0\%$) to highly shared ($S = 83.3\%$). This allows us to identify the optimal ratio that minimizes training loss for a given computational budget. The problem is formalized as:

$$S^{\text{opt}} = \arg \min_S \mathcal{L}(S; C, M, A, G) \tag{9}$$

where $S^{\text{opt}}$ is the optimal shared expert that minimizes the training loss $\mathcal{L}$ under a fixed FLOPs budget $C$, model size $M$, activation ratio $A$, and granularity $G$. Our experiments, as depicted in Figure 13a, reveal a U-shaped relationship between the shared expert ratio and training loss. The minimum loss is generally achieved at a relatively low shared expert ratio, while having no shared experts ($S = 0\%$) usually results in suboptimal performance. Furthermore, we observe a subtle trend where the optimal sharing ratio appears to scale with the compute budget. This is supported by our empirical scaling law (EL) analysis in Figure 13b, which shows that lower FLOPs budgets

($\leq 10^{20}$) benefit from a slightly higher sharing ratio ($S = 16.7\%$), whereas larger budgets ($> 10^{20}$) achieve greater efficiency with a lower ratio ($S = 8.3\%$).

Since large-scale pre-training runs typically exceed $10^{20}$ FLOPs, this suggests a practical heuristic: the optimal design choice is to use the lowest possible non-zero sharing ratio. Assuming the dimensions of shared and regular experts are equal, this can be heuristically implemented by setting the number of shared experts to one.

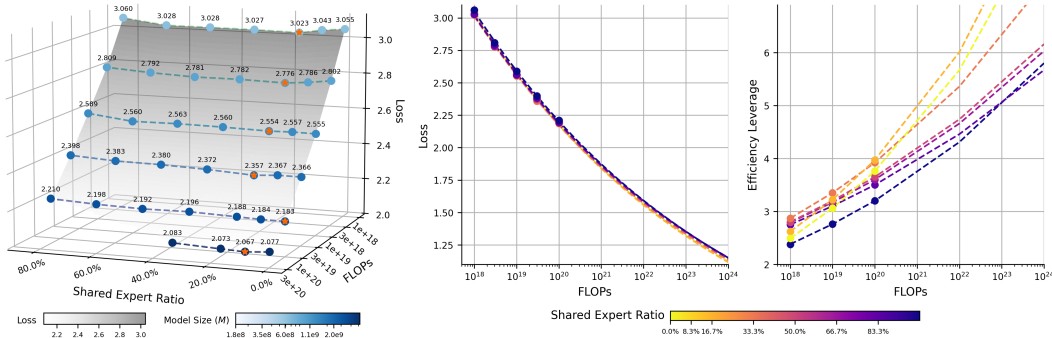

(a) IsoFLOPs curves over varying $G$.          (b) Loss and efficiency leverage scaling curve over varying $G$.

Figure 13: Impact of the Shared Ratio $S$ on Loss and Efficiency.    (a) Loss curves demonstrate that a low, non-zero sharing ratio minimizes training loss, outperforming both no shared experts ($S = 0\%$) and highly shared configurations.. (b) EL analysis reveal that the optimal sharing ratio is higher ($S = 16.7\%$) for smaller FLOPs ($< 10^{20}$) and decreases to $S = 8.3\%$ for larger FLOPs ($> 10^{20}$).

---

**Key Takeaway 3**

- **Optimal Sharing Ratio Exhibits a Subtle Scaling Trend.** We identify a subtle scaling trend between the optimal shared expert ratio and the compute budget: the ideal ratio decreases as the compute budget increases.

- **"One Shared Expert" Rule for Large-Scale Training.** For large-scale pre-training with uniformly sized experts, the optimal design heuristic is to employ a single shared expert. This configuration establishes the minimal non-zero sharing ratio.

---

### F.4 OTHER CONFIGURATIONS OF MOE ARCHITECTURE

**Arrangement of MoE and Dense Layers**    To ensure balanced routing in the early layers, mainstream MoE models typically replace all FFNs except for the first few layers with MoE layers. We investigate the impact of this design decision on the efficiency of MoE models. To ensure a meaningful exploration space, we extend all models in our experiments to 60 layers and set the first 1, 2, or 3 layers as dense layers sequentially. The dimension of these dense layers is set to match the total dimension of the activated experts in the corresponding MoE layers, ensuring the overall computational cost (FLOPs/token) remains constant. This design allows us to isolate and study the effect of the proportion of dense layers on MoE efficiency. The experimental results, presented in Figure 14a and 14b, reveal the following key findings: 1) From a model performance perspective, replacing the first few layers with dense layers has a minor impact. Using a dense proportion of zero as the baseline, we estimated the efficiency leverage for each configuration. Within a FLOPs budget of up to $1 \times 10^{24}$ FLOPs, the efficiency leverage remains close to 1. This indicates that configuring the initial layers as dense offers negligible efficiency improvement. However, this adjustment effectively reduces the total number of parameters in the model and mitigates routing imbalances in the early layers. Thus, despite its limited efficiency gains, this remains a valuable design optimization. 2) Further investigation into the optimal proportion of dense layers under varying computational budgets reveals a trend: as FLOPs budgets increase, the optimal dense proportion also grows. For example, in our experiments, when the compute budget is $1 \times 10^{18}$ FLOPs, the optimal dense proportion is

zero. As the compute budget increases to $3 \times 10^{20}$ FLOPs, the optimal dense layer proportion shifts to approximately $2/60$ or $3/60$.

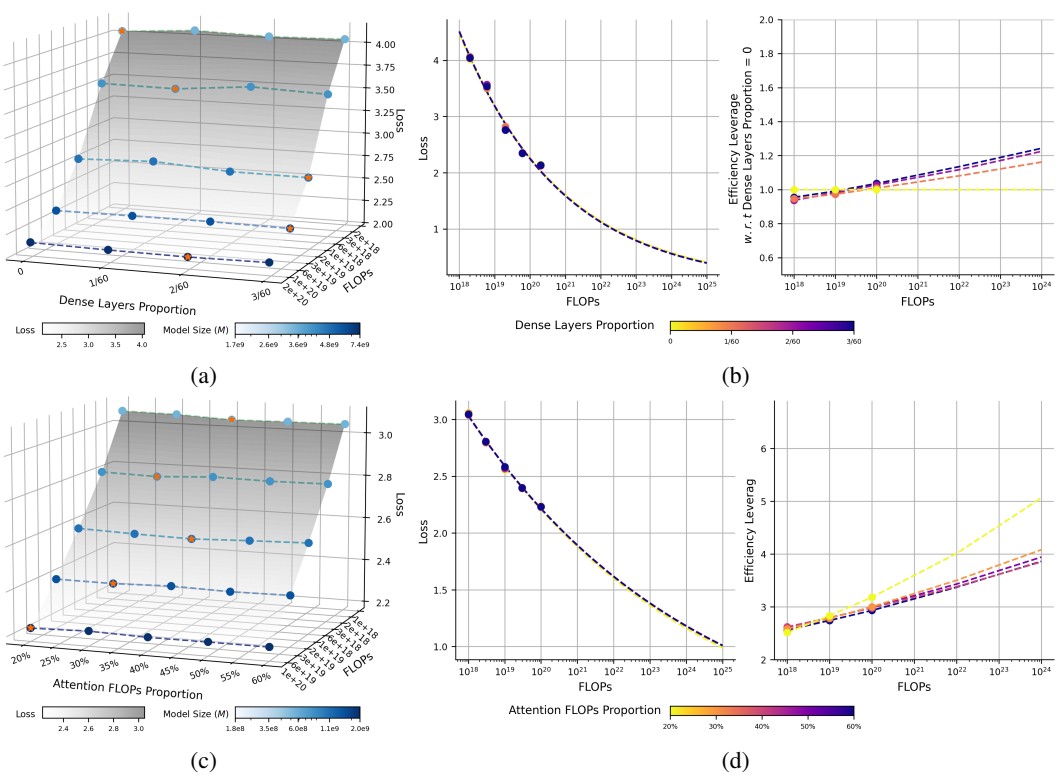

Figure 14: Impact of Dense Layers Proportion and Compute Budget Allocation between Attention and FFN. (a,b) Replacing the first few layers with dense layers shows minor impact on model performance. As computational budgets increase, the optimal proportion of dense layers also gradually rises. (c,d) Modifying the attention FLOPs ratio within a broad range (20%-50%) has a negligible influence on model performance, demonstrating the robustness of this configuration.

**Compute Resource Allocation between Attention and FFN**   As two core components of the Transformer model, the attention mechanism (Attention) and FFN account for the majority of the model's computational load. To this end, we explore the impact of computational allocation between the attention mechanism and the FFN on the efficiency of the MoE model. Specifically, we construct a series of models with fixed model scale $M$ but varying compute budgets by increasing the hidden layer size of the attention module while reducing the hidden layer size of each expert in the MoE. We then observe the performance changes of these models under different computational allocations and evaluate their scaling trends. The experimental results are illustrated in Figure 14c and 14d, revealing the following key findings: 1) When the attention FLOPs ratio is between 30% and 40%, it represents a relatively stable and reliable configuration. Models tend to achieve optimal or near-optimal performance within this range. This configuration is consistent with the default settings of mainstream open-source MoE models. 2) Adjusting the attention FLOPs ratio within a broader range (20%-50%) has minor impact on model performance. As shown in Figure 14d, the loss scaling curves and efficiency leverage of these models are nearly identical. Since the attention mechanism generally has a higher computational density (*i.e.,* FLOPs-per-parameter) compared to the FFN, increasing the attention FLOPs ratio while keeping the overall model size constant reduces the total number of model parameters, resulting in higher knowledge density. However, this also implies potentially higher downstream inference costs.

> **Key Takeaway 4**
>
> - **Introducing Dense Layers is a Valuable Design Optimization.** Incorporating dense layers in the early stages of MoE has minor impact on efficiency but helps mitigate routing imbalances and reduces overall parameters. The optimal proportion of dense layers increases with higher FLOPs budgets, though it offers limited efficiency gains.
>
> - **Robustness of Compute Budget Allocation between Attention and FFN** Allocating 30%-40% of FLOPs to the attention mechanism achieves optimal or near-optimal performance, with minor impact outside this range. Increasing attention FLOPs proportion enhances knowledge density but reduces downstream inference efficiency.

## G    Values of the Fitted Coefficients.

To validate the proposed scaling law for EL, we fit Eq. 4 using Huber loss and the BFGS optimization algorithm (Hoffmann et al., 2022). We use data points with an EL factor below 6 for training, while those are reserved as a validation set. The values are presented in Appendix 6.

Table 6: **Values of the Fitted Coefficients.**

| $a$ | $d$ | $\gamma$ | $\beta$ | $A_{start}$ | $A_{max}$ |
|------|---------|---------|----------|-------------|-----------|
| 1.23 | -7.61e-2 | 1.67e-2 | -1.17e-1 | 1.63e-2 | 5.28e+16 |

## H    Detailed Results of MoE-mini Evaluation

**Evaluation Benchmarks**    To evaluate performance, we consider a diverse suite of downstream tasks designed to provide a holistic assessment of model capabilities. These tasks are grouped into several categories, such as: (a) General Knowledge/Reasoning (*e.g.,* ARC (Bhakthavatsalam et al., 2021), AGIEval (Zhong et al., 2024), OpenBookQA (Mihaylov et al., 2018), BBH (Suzgun et al., 2023), ProntoQA (Saparov & He, 2023), PIQA (Bisk et al., 2020), HellaSwag (Zellers et al., 2019), Multi-LogiEval (Patel et al., 2024)) (b) Language Understanding (*e.g.,* RACE (Lai et al., 2017)) (c) Professional Knowledge (*e.g.,* MMLU (Hendrycks et al., 2021a), CMMLU (Li et al., 2024), MMLU-Pro (Wang et al., 2024b), GPQA (Rein et al., 2023), C-Eval (Huang et al., 2023), CommonsenseQA (Talmor et al., 2018)) (d) Math (*e.g.,* GSM8K (Cobbe et al., 2021), MATH (Hendrycks et al., 2021b), GAOKAO (Zhang et al., 2023), Gaokao2023-Math-En, MGSM (Shi et al., 2023), CMATH (Wei et al., 2023), MathBench (Liu et al., 2024), Minerva-Math (Lewkowycz et al., 2022), CN-Middle School 24) (e) Code (*e.g.,* Humaneval (Chen et al., 2021), HumanEval-cn (Peng et al., 2024), HumanEval-plus (Liu et al., 2023), HumanEval-FIM (Bavarian et al., 2022), Live-CodeBench (Jain et al., 2025), MBPP (Tao et al., 2024), MBPP-Plus (Liu et al., 2023), CruxEval (Gu et al., 2024)).

**Evaluation Results**    The comparative evaluation in Table 7 reveals that MoE-mini achieves an average score of 45.5, surpassing Dense-6.1B's 44.0. This result compellingly demonstrates that MoE-mini accomplishes a "small yet powerful" feat with significantly lower inference costs, its activated parameters amount to only about *13%* of its competitor's, striking an exceptional balance between performance and efficiency.

Upon closer examination of performance across specific dimensions, MoE-mini's advantages are both comprehensive and focused. In general knowledge and reasoning tasks, it exhibits notable advantages in open-ended question answering tasks such as OpenBookQA and complex logical reasoning benchmarks like Multi-LogiEval. This trend continues in specialized knowledge domains, where MoE-mini delivers better results on comprehensive academic benchmarks like MMLU and MMLU-Pro. Its superiority is particularly evident in language understanding tasks, as it consistently outperforms its competitor in the RACE series of reading comprehension tests, showcasing stronger contextual understanding capabilities. In tasks requiring high coding proficiency, MoE-mini stands out significantly, especially in the HumanEval-Plus benchmark, which measures code robustness,

Table 7: Detailed performance comparison of `MoE-mini` (17B-A0.8B) and `Dense-6.1B`.

| | Metric | Dense-6.1B | MoE-mini (A0.8B) |
|---|---|---|---|
| | ARC-challenge | 59.7 | 57.0 |
| | ARC-easy | 78.0 | **78.7** |
| | AGIEval | 33.4 | **34.9** |
| | OpenBookQA | 68.6 | **75.2** |
| General Knowledge | BBH | **48.0** | 35.7 |
| /Reasoning | ProntoQA | 16.5 | **19.5** |
| | Multi-LogiEval | 55.6 | **61.3** |
| | HellaSwag | 65.6 | **66.6** |
| | PIQA | 76.6 | **77.2** |
| | Average | 55.8 | **56.2** |
| | MMLU | 51.1 | **53.1** |
| | MMLU-Pro | 21.7 | **24.0** |
| Professional | CMMLU | 50.7 | **51.9** |
| Knowledge | C-Eval | **52.5** | 51.1 |
| | CommonsenseQA | **63.6** | 60.6 |
| | GPQA | 24.8 | **27.3** |
| | Average | 44.0 | **44.7** |
| Language | RACE-middle | 73.4 | **75.6** |
| Understanding | RACE-high | 65.0 | **67.6** |
| | Average | 69.2 | **71.6** |
| | HumanEval | 31.7 | **35.4** |
| | HumanEval-cn | **34.2** | 32.3 |
| | HumanEval-Plus | 35.4 | **51.8** |
| | HumanEval-FIM | **62.8** | 61.3 |
| Code | MBPP | 41.0 | **44.6** |
| | MBPP-Plus | 50.0 | **51.6** |
| | LiveCodeBench | **7.5** | 7.4 |
| | CruxEval | 32.9 | **34.1** |
| | Average | 36.9 | **39.8** |
| | GSM8K | **59.2** | 58.0 |
| | MATH | 23.7 | **29.8** |
| | CMATH | 60.5 | **62.9** |
| | MGSM-zh | 35.6 | **36.8** |
| | CN-Middle School 24 | 41.6 | **42.6** |
| Math | Minerva-Math | **3.3** | 2.9 |
| | MathBench | 27.5 | **28.6** |
| | Gaokao2023-Math-En | 33.1 | **33.5** |
| | GAOKAO-Math24 | 12.1 | **17.6** |
| | Average | 32.9 | **34.7** |
| Overall Average | | 44.0 | **45.5** |

achieving an impressive lead of over *16 points*. Similarly, in mathematical reasoning, while slightly lagging in basic arithmetic tasks like GSM8K, it excels in challenging benchmarks such as MATH and GAOKAO-Math24, demonstrating strong potential in solving complex problems. Collectively, `MoE-mini` achieves a 1.5-point overall advantage, validating its parameter-efficient MoE design. It not only drastically reduces inference costs through sparse activation but, more critically, its "expert networks" seem to enable higher performance ceilings in key areas such as language understanding, code generation, and advanced reasoning.

**Pre-training Evaluation of MoE-mini**    We present a detailed evaluation of `MoE-mini`'s training process. Figure 15 provides a comprehensive comparison across datasets and categories, as outlined in the main experiments in Section 5.2. The results show that `MoE-mini` achieves comparable performance to `Dense-6.1B` on the majority of datasets.

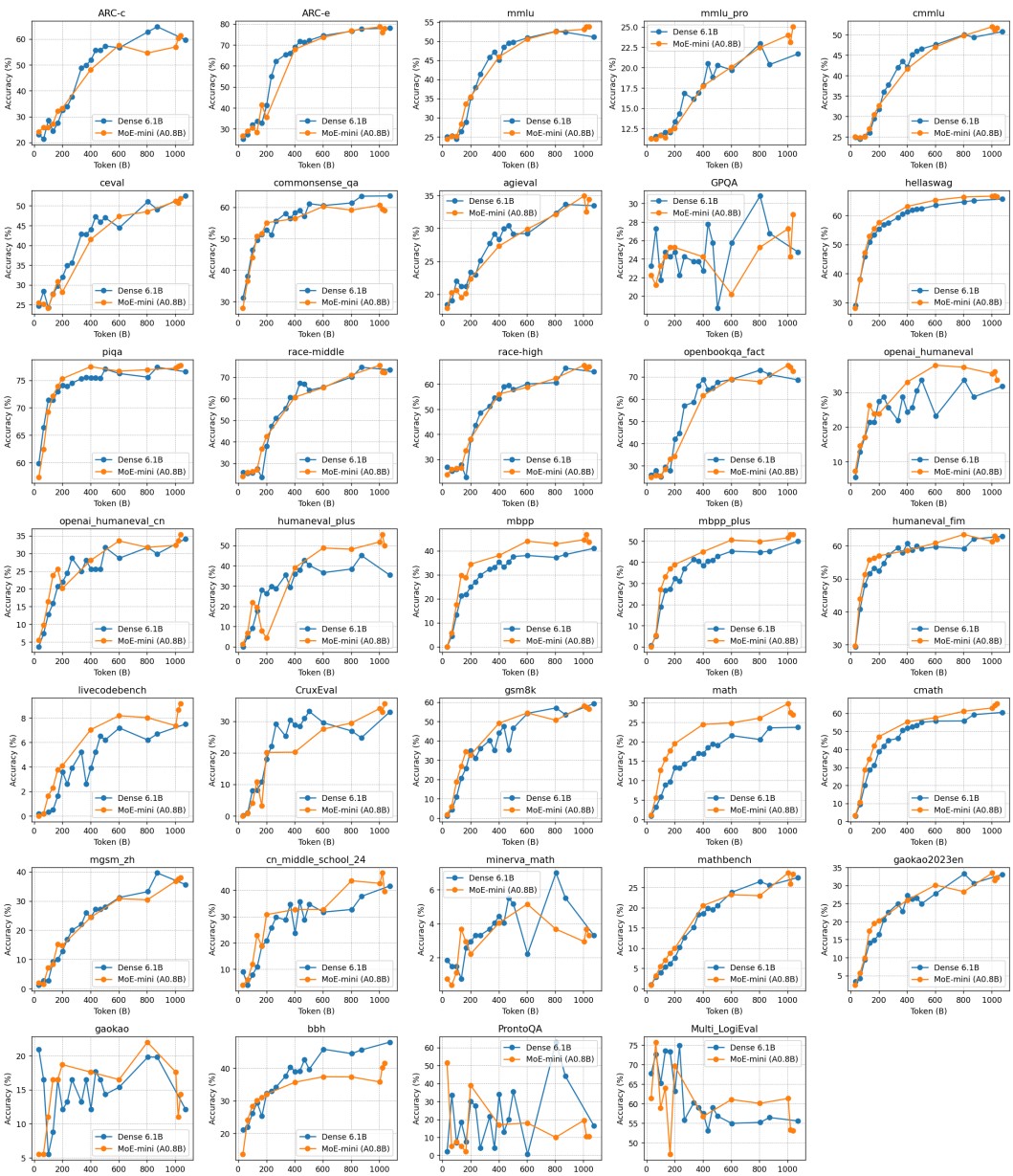

Figure 15: Overall and category-wise performance comparison between MoE-mini (17B-A0.8B) and Dense-6.1B.

## I    ESTIMATING FLOPs

To analyze the efficiency of our models, we quantify the computational cost in terms of total training Floating Point Operations (FLOPs). Following standard practice (Kaplan et al., 2020), we estimate the total training FLOPs as approximately three times the cost of a single forward pass ($C_{\text{train}} \approx 3 \cdot C_{\text{fwd}}$). The forward pass FLOPs are the sum of computations from the attention and feed-forward network (FFN) layers, plus a final logit projection.

For a model with hidden size $d_{\text{model}}$, batch size $B$, and sequence length $s$, the cost of the attention block per layer, $C_{\text{attn}}$, which includes Grouped-Query Attention (GQA) (Ainslie et al., 2023) and all projections, is approximately:

$$C_{\text{attn}} \approx 2Bsd_{\text{model}}^2 \left(1 + \frac{2}{n_h/n_{kv}}\right) + 4Bs^2 d_{\text{model}} \tag{10}$$

where $n_h$ and $n_{kv}$ are the number of attention and key-value heads, respectively. The FFN cost varies by layer type. A dense layer with intermediate size $d_{\text{ffn}}$ requires $C_{\text{dense\_ffn}} = 6Bsd_{\text{model}}d_{\text{ffn}}$ FLOPs. A MoE layer activating $E_a$ experts, each with size $d_{\text{expert}}$, requires:

$$C_{\text{moe\_ffn}} \approx 6Bsd_{\text{model}}(E_a \cdot d_{\text{expert}}) \tag{11}$$

If a shared expert of size $d_{\text{shared}}$ is used, its cost, $4Bsd_{\text{model}}d_{\text{shared}}$, is added. For a model with $L$ layers (of which the first $L_{\text{dense}}$ are dense) and a vocabulary of size $V$, the total forward FLOPs are:

$$C_{\text{fwd}} = \sum_{i=1}^{L}(C_{\text{attn}} + C_{\text{ffn},i}) + 2Bsd_{\text{model}}V \tag{12}$$

where $C_{\text{ffn},i}$ is the FFN cost for the $i$-th layer, which can be either $C_{\text{dense\_ffn}}$ or $C_{\text{moe\_ffn}}$.

## J    LIST OF EXPERIMENTAL MODELS

The detailed configurations for all experiments conducted in this study are presented in Tables 8 (activation ratio), Tables 9 (expert granularity), Tables 10 (shared experts), Tables 11 (layer arrangement), and Tables 12 (compute allocation between attention and FFNs).

## K    METHODOLOGY FOR CALCULATING EFFECTIVE LEVERAGE (EL)

Our methodology for obtaining EL datapoints does not involve training a unique dense model to match the loss of each individual MoE run. Instead, we employ a more systematic and scalable approach based on modeling the loss-compute scaling behavior for each model family. For a specific MoE architecture (e.g., for a fixed activation ratio and granularity), as show in Algorithm 1, the process of obtaining data points for EL is as follows:

**1. Collecting (compute budget, optimal loss) Data:**    We first train a suite of MoE models (see Tables 8 to 12) and a corresponding suite of dense counterparts. For a given MoE configuration, its dense counterpart is defined as a standard Transformer architecture, equivalent to an MoE model with a 100% activation rate. **As shown in Table 8, the model architecture with an activation rate of 1.0 serves as the dense counterpart for all other MoE configurations in our study.** All models are trained on the same dataset with the same recipe, for up to $3 \times 10^{20}$ FLOPs. This process generates a set of (compute, optimal loss) data points $\{(C, \ell)\}$ for both the specific MoE architecture $\mathcal{X}_{\text{MoE}}$ and dense architecture $\mathcal{X}_{\text{Dense}}$. This is illustrated in Figure 10a, Figure 11a, and Figure 13a.

**2. Fitting Loss Scaling Curves:**    We then fit separate loss scaling functions, $L_{\mathcal{X}_{\text{MoE}}}(\cdot)$ and $L_{\mathcal{X}_{\text{Dense}}}(\cdot)$, to the collected data for the specific MoE and dense architecture, respectively. We use a standard power-law form, $L_{\mathcal{X}}(C) = \alpha_{\mathcal{X}} C^{\beta_{\mathcal{X}}} + b_{\mathcal{X}}$, consistent with prior work (Kaplan et al., 2020; Henighan et al., 2020; Achiam et al., 2023). This process yields smooth loss scaling functions that can predict the architecture's optimal loss at any given compute budget $C$. These fitted curves are shown in the left panels of Figure 10b, Figure 11b, and Figure 13b.

**3. Computing EL via Interpolated Loss Matching**  For an MoE model at a compute budget $C_{\text{MoE}}$, we first use its fitted curve to calculate the predicted loss: $\mathcal{L}^{\star} = L_{\mathcal{X}_{\text{MoE}}}(C_{\text{MoE}})$. Next, we use the dense model's loss curve, $L_{\mathcal{X}_{\text{Dense}}}(C)$, to find the compute required to achieve the same loss by solving $C_{\text{Dense}} = L_{\mathcal{X}_{\text{Dense}}}^{-1}(\mathcal{L}^{\star})$. Finally, we compute the EL as defined: $\text{EL} = C_{\text{dense}}/C_{\text{MoE}}$. These EL values are shown in the right panels of Figure 10b, Figure 11b, and Figure 13b. This methodology based loss scaling curves on allows us to systematically evaluate EL across a continuous range of compute budgets.

---

**Algorithm 1** Calculating Efficiency Leverage (EL)

---

**Require:** $P_{\text{dense}}, P_{\text{MoE}}$: Sets of (compute, loss) data points from dense and MoE training runs.
**Require:** $C_{\text{MoE}}$: The target MoE compute budget for which to calculate EL.
**Ensure:** EL: The calculated Efficiency Leverage value.

1:               ▷ **Part 1: Fit Loss Scaling Functions from Data**
2: **for** each model family $\mathcal{X} \in \{\text{Dense}, \text{MoE}\}$ **do**
3:    Fit a continuous loss scaling function $L_{\mathcal{X}}(C)$ to the corresponding data points $P_{\mathcal{X}}$.
       ▷ Typically uses a parametric model like a power law, e.g., $L(C) = aC^{-b} + c$.
4: **end for**

5:                ▷ **Part 2: Calculate EL via Loss Matching**
6: $\mathcal{L}^{\star} \leftarrow L_{\mathcal{X}_{\text{MoE}}}(C_{\text{MoE}})$      ▷ Calculate the target loss achieved by the MoE model.
7: **if** $\mathcal{L}^{\star}$ is not attainable by the dense model (e.g., below its loss floor) **then**
8:    **return undefined**
9: **end if**
10: Solve for $C_{\mathcal{X}_{\text{Dense}}}$ such that $L_{\text{Dense}}(C_{\text{Dense}}) = \mathcal{L}^{\star}$.
     ▷ This is done by inverting the parametric function or using a numerical root-finder.
11: $\text{EL} \leftarrow \frac{C_{\text{Dense}}}{C_{\text{MoE}}}$        ▷ The final EL is the ratio of compute budgets.
12: **return** EL

---

## L Computational Resources

Our study utilized a total of approximately **680,000 equivalent H800 GPU-hours**. The allocation of this computational budget is detailed below:

- A total of **360,000 hours** were used for preliminary experiments. This phase involved extensive hyperparameter tuning and explorations into optimal model and data allocation.

- The main architectural scaling experiments, which form the core contribution of this work, required **200,000 hours**.

- Final validation runs, including the complete training of our 16-billion parameter MoE and dense models on 1T tokens, consumed the remaining **120,000 hours**.

This substantial investment underpins the reliability and scale of our empirical findings.

## M Impact of the Number of Attention Heads

To investigate the impact of the number of attention heads ($n_{\text{head}}$), we conducted a series of supplementary experiments. For each model size, we systematically varied $n_{\text{head}}$ while keeping all other hyperparameters constant, as detailed in Table 14. The results, visualized in Figure 16, reveal that model performance is not sensitive to a single specific number of heads. Instead, we identified a range of "near-optimal" values, defined as configurations achieving a final validation loss within 0.5% of the minimum observed loss for that model size. Based on this observation, and in line with common practice in scaling law studies (Hoffmann et al., 2022; Ludziejewski et al., 2024), we scale $n_{\text{head}}$ proportionally with the model dimension ($d_{\text{model}}$) in our main experiments (see Tables 8 to 12). This approach ensures a robust and fair comparison across different model scales.

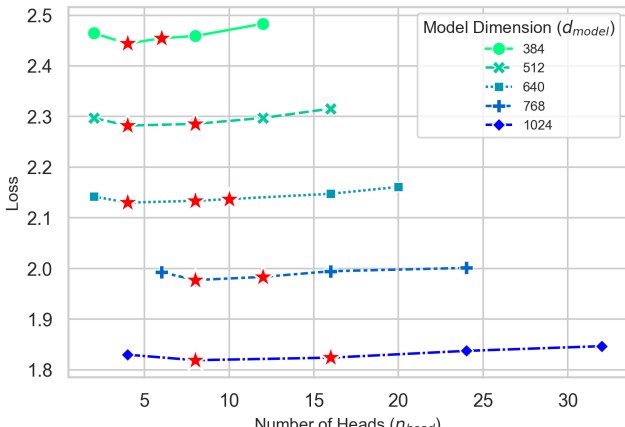

Figure 16: Impact of the Number of Attention Heads. Red stars denote "near-optimal" configurations, defined as those achieving a final loss within 0.5% of the minimum for each model size.

# N  GOODNESS-OF-FIT ANALYSIS FOR SCALING LAWS

Our methodology is empirical, aligning with established practices in scaling law research (Clark et al., 2022; Hoffmann et al., 2022; Ludziejewski et al., 2024; Abnar et al., 2025; Besiroglu et al., 2024). For each component of the scaling law, we first identified the underlying trend through data visualization. We then selected a functional form that best captures this trend and rigorously validated our choice by comparing its goodness-of-fit (e.g., $R^2$), against simpler alternatives.

## N.1  SATURATING TRANSFORM FOR ACTIVATION RATIO ($A$)

We empirically observed a trend of diminishing returns when decreasing the activation ratio ($A$); that is, the performance gains from a smaller ratio lessen as $A$ approaches zero. A saturating transformation, which has been successfully employed to model similar phenomena (Clark et al., 2022), is well-suited to capture this effect. To validate this choice, we compared the fit of our proposed form (Eq. 1) against a standard power-law model. As shown in Table 16, the saturating transformation achieves a significantly higher R-squared value, confirming its superior fit to the data.

## N.2  LOG-POLYNOMIAL FUNCTION FOR EXPERT GRANULARITY ($G$)

Our analysis revealed a U-shaped relationship between expert granularity ($G$) and model performance, where the EL first increases and then decreases as $G$ grows. A quadratic polynomial function is a standard and parsimonious choice for modeling such non-monotonic trends and is widely used in prior work (Ludziejewski et al., 2024; Abnar et al., 2025). We confirmed its suitability by comparing its fit against a standard power-law and a saturating transform. The results in Table 17 demonstrate that the log-polynomial form provides the most accurate fit to our empirical observations.

## N.3  POWER LAW FOR COMPUTE BUDGET ($C$)

The selection of a power law for the compute budget ($C$) was directly motivated by our empirical data. As visualized in Figure 4c, the log-log plot of Effective Loss versus training FLOPs exhibits a distinct linear relationship. This linearity is the hallmark of a power-law dependency, making it the natural and most appropriate functional form for this component of our model.

## N.4  JOINT SCALING LAW FOR EFFICIENCY LEVERAGE

Finally, the joint scaling law (Eq. 4) is an empirical composite model designed to synthesize our individual findings. Its structure is not arbitrary but reflects the observed interactions: The term for Activation Ratio ($A$) serves as the primary driver of efficiency. This is then modulated by the independent, non-linear adjustment from Granularity ($G$). Finally, the entire efficiency gain is amplified

by the Compute Budget ($C$) through the overarching power-law pattern. This composite structure provides the most comprehensive explanation of the joint effects. Its high accuracy is empirically validated in Figure 5, achieving an R-squared of 0.9858. The model demonstrates strong predictive power with a low RMSE on both the training set (0.2169 over 200 points) and the validation set (0.5275 over 24 points). We further analyzed the residuals of the fitted scaling law, as shown in Figure 17. The residuals are approximately normally distributed and centered at zero (mean = -0.0273, std. dev. = 0.2803), which confirms the high quality of the fit.

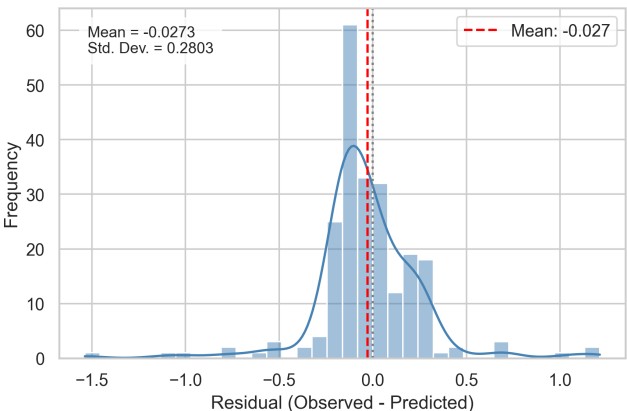

Figure 17: Plot of residuals of our estimated joint scaling law.

Table 8: Experimental configurations for the expert activation ratio analysis. Within each group, the number of activated experts ($E_a = 2$) is fixed, while the total number of experts ($E$) is varied to study the effect of the activation ratio.

| $n_{layers}$ | $d_{model}$ | $d_{expert}$ | $n_{heads}$ | $n_{kv\_head}$ | $E$ | $E_s$ | $\eta$ | $B$ | Max training FLOPs |
|---|---|---|---|---|---|---|---|---|---|
| 8 | 384 | 320 | 8 | 2 | [2,4,8,16,32,64,128,256] | 1 | 1.52e-3 | 98 | 2e18 |
| 8 | 512 | 512 | 8 | 2 | [2,4,8,16,32,64,128,256] | 1 | 1.31e-3 | 147 | 6e18 |
| 10 | 640 | 640 | 10 | 2 | [2,4,8,16,32,64,128,256] | 1 | 1.11e-3 | 228 | 2e19 |
| 14 | 768 | 768 | 12 | 4 | [2,4,8,16,32,64,128,256] | 1 | 9.5e-4 | 342 | 6e19 |
| 16 | 1024 | 1024 | 16 | 4 | [2,4,8,16,32,64,128,256] | 1 | 8.1e-4 | 531 | 2e20 |
| 22 | 1280 | 1280 | 20 | 4 | [2,4,8,16,32,64,128,256] | 1 | 7.0e-4 | 795 | 6e20 |

Table 9: Experimental configurations for the expert granularity analysis. Within each group, the base model architecture is fixed while the MoE configuration (total experts $E$, activated experts $E_a$, shared experts $E_s$, and expert dimension $d_{\text{expert}}$) is varied to study the effect of granularity.

| $n_{layers}$ | $d_{model}$ | $n_{\text{heads}}$ | $E$ | $E_a$ | $E_s$ | $d_{\text{expert}}$ | $B$ | $\eta$ | Max training FLOPs |
|---|---|---|---|---|---|---|---|---|---|
| 8 | 384 | 8 | 64 | 2 | 1 | 384 | 98 | 1.52e-3 | 2e18 |
| | | | 128 | 4 | 2 | 192 | | | |
| | | | 192 | 6 | 3 | 128 | | | |
| | | | 256 | 8 | 4 | 96 | | | |
| | | | 384 | 12 | 6 | 64 | | | |
| | | | 512 | 16 | 8 | 48 | | | |
| 8 | 512 | 8 | 64 | 2 | 1 | 512 | 147 | 1.31e-3 | 6e18 |
| | | | 128 | 4 | 2 | 256 | | | |
| | | | 192 | 6 | 3 | 170 | | | |
| | | | 256 | 8 | 4 | 128 | | | |
| | | | 384 | 12 | 6 | 85 | | | |
| | | | 512 | 16 | 8 | 64 | | | |
| 10 | 640 | 10 | 64 | 2 | 1 | 640 | 228 | 1.11e-3 | 2e19 |
| | | | 128 | 4 | 2 | 320 | | | |
| | | | 192 | 6 | 3 | 213 | | | |
| | | | 256 | 8 | 4 | 160 | | | |
| | | | 384 | 12 | 6 | 106 | | | |
| | | | 512 | 16 | 8 | 80 | | | |
| 14 | 768 | 12 | 64 | 2 | 1 | 768 | 342 | 9.5e-4 | 6e19 |
| | | | 128 | 4 | 2 | 384 | | | |
| | | | 192 | 6 | 3 | 256 | | | |
| | | | 256 | 8 | 4 | 192 | | | |
| | | | 384 | 12 | 6 | 128 | | | |
| | | | 512 | 16 | 8 | 96 | | | |
| 16 | 1024 | 16 | 64 | 2 | 1 | 1024 | 531 | 8.1e-4 | 2e20 |
| | | | 128 | 4 | 2 | 512 | | | |
| | | | 192 | 6 | 3 | 341 | | | |
| | | | 256 | 8 | 4 | 256 | | | |
| | | | 384 | 12 | 6 | 170 | | | |
| | | | 512 | 16 | 8 | 128 | | | |
| 22 | 1280 | 20 | 64 | 2 | 1 | 1280 | 795 | 7.0e-4 | 6e20 |
| | | | 128 | 4 | 2 | 640 | | | |
| | | | 192 | 6 | 3 | 426 | | | |
| | | | 256 | 8 | 4 | 320 | | | |
| | | | 384 | 12 | 6 | 213 | | | |
| | | | 512 | 16 | 8 | 160 | | | |

Table 10: Experimental configurations for the shared expert ratio analysis. Within each group, we fix the total number of experts ($E = 256$) and the total number of activated pathways ($E_a + E_s = 12$), while varying the ratio between specialized experts ($E_a$) and shared experts ($E_s$) to study its impact on performance.

| $nlayers$ | $d_{model}$ | $n_{heads}$ | $E$ | $E_a$ | $E_s$ | $d_{expert}$ | $B$ | $\eta$ | Max training FLOPs |
|---|---|---|---|---|---|---|---|---|---|
| 8 | 384 | 8 | 256 | 2 | 10 | 96 | 98 | 1.52e-3 | 2e18 |
|  |  |  | 256 | 4 | 8 | 96 |  |  |  |
|  |  |  | 256 | 6 | 6 | 96 |  |  |  |
|  |  |  | 256 | 8 | 4 | 96 |  |  |  |
|  |  |  | 256 | 11 | 1 | 96 |  |  |  |
|  |  |  | 256 | 12 | 0 | 96 |  |  |  |
| 8 | 512 | 8 | 256 | 2 | 10 | 128 | 147 | 1.31e-3 | 6e18 |
|  |  |  | 256 | 4 | 8 | 128 |  |  |  |
|  |  |  | 256 | 6 | 6 | 128 |  |  |  |
|  |  |  | 256 | 8 | 4 | 128 |  |  |  |
|  |  |  | 256 | 11 | 1 | 128 |  |  |  |
|  |  |  | 256 | 12 | 0 | 128 |  |  |  |
| 10 | 640 | 10 | 256 | 2 | 10 | 160 | 228 | 1.11e-3 | 2e19 |
|  |  |  | 256 | 4 | 8 | 160 |  |  |  |
|  |  |  | 256 | 6 | 6 | 160 |  |  |  |
|  |  |  | 256 | 8 | 4 | 160 |  |  |  |
|  |  |  | 256 | 11 | 1 | 160 |  |  |  |
|  |  |  | 256 | 12 | 0 | 160 |  |  |  |
| 14 | 768 | 12 | 256 | 2 | 10 | 192 | 342 | 9.5e-4 | 6e19 |
|  |  |  | 256 | 4 | 8 | 192 |  |  |  |
|  |  |  | 256 | 6 | 6 | 192 |  |  |  |
|  |  |  | 256 | 8 | 4 | 192 |  |  |  |
|  |  |  | 256 | 11 | 2 | 192 |  |  |  |
|  |  |  | 256 | 12 | 0 | 192 |  |  |  |
| 16 | 1024 | 16 | 256 | 2 | 10 | 256 | 531 | 8.1e-4 | 2e20 |
|  |  |  | 256 | 4 | 8 | 256 |  |  |  |
|  |  |  | 256 | 6 | 6 | 256 |  |  |  |
|  |  |  | 256 | 8 | 4 | 256 |  |  |  |
|  |  |  | 256 | 11 | 1 | 256 |  |  |  |
|  |  |  | 256 | 12 | 0 | 256 |  |  |  |
| 22 | 1280 | 20 | 256 | 2 | 10 | 320 | 795 | 7.0e-4 | 6e20 |
|  |  |  | 256 | 4 | 8 | 320 |  |  |  |
|  |  |  | 256 | 6 | 6 | 320 |  |  |  |
|  |  |  | 256 | 8 | 4 | 320 |  |  |  |
|  |  |  | 256 | 11 | 1 | 320 |  |  |  |
|  |  |  | 256 | 12 | 0 | 320 |  |  |  |

Table 11: Experimental configurations for the arrangement of MoE and dense layers analysis. Within each group, the total number of layers is fixed at 60, while the mix of dense layers ($n_{\text{dense\_layers}}$) and MoE layers ($n_{\text{moe\_layers}}$) is varied to study the impact of their ratio and placement on performance.

| $n_{layers}$ | $n_{dense\_layers}$ | $n_{moe\_layers}$ | $d_{model}$ | $d_{ffn}$ | $n_{\text{heads}}$ | $E$ | $E_a$ | $E_s$ | $d_{\text{expert}}$ | $B$ | $\eta$ | Max training FLOPs |
|---|---|---|---|---|---|---|---|---|---|---|---|---|
| 60 | 0
1
2
3 | 60
59
58
57 | 384 | 1280 | 8 | 64 | 2 | 1 | 384 | 98 | 1.52e-3 | 2e18 |
| 60 | 0
1
2
3 | 60
59
58
57 | 512 | 2048 | 8 | 64 | 2 | 1 | 512 | 147 | 1.31e-3 | 6e18 |
| 60 | 0
1
2
3 | 60
59
58
57 | 640 | 2560 | 10 | 64 | 2 | 1 | 640 | 228 | 1.11e-3 | 2e19 |
| 60 | 0
1
2
3 | 60
59
58
57 | 768 | 3072 | 12 | 64 | 2 | 1 | 768 | 342 | 9.5e-4 | 6e19 |
| 60 | 0
1
2
3 | 60
59
58
57 | 1024 | 4096 | 16 | 64 | 2 | 1 | 1024 | 531 | 8.1e-4 | 2e20 |
| 60 | 0
1
2
3 | 60
59
58
57 | 1280 | 5120 | 20 | 64 | 2 | 1 | 1280 | 795 | 7.0e-4 | 6e20 |

Table 12: Experimental configurations for analyzing the compute allocation between attention and FFNs. Within each group, the core MoE structure is held constant, while we systematically vary the model's hidden dimension ($d_{\text{model}}$) and the expert dimension ($d_{\text{expert}}$) to explore the optimal trade-off in compute allocation between the attention mechanism and the FFN experts.

| $layers$ | $d_{model}$ | $d_{expert}$ | $n_{heads}$ | $n_{kv\_head}$ | $E$ | $E_s$ | $E_a$ | $\eta$ | $B$ | Max training FLOPs |
|---|---|---|---|---|---|---|---|---|---|---|
| 8 | 352 | 450 | 8 | 2 | 64 | 1 | 2 | 1.52e-3 | 96 | 2e18 |
| 8 | 368 | 380 | 8 | 2 | 64 | 1 | 2 | 1.52e-3 | 96 | 2e18 |
| 8 | 384 | 320 | 8 | 2 | 64 | 1 | 2 | 1.52e-3 | 96 | 2e18 |
| 8 | 400 | 260 | 8 | 2 | 64 | 1 | 2 | 1.52e-3 | 96 | 2e18 |
| 8 | 416 | 208 | 8 | 2 | 64 | 1 | 2 | 1.52e-3 | 96 | 2e18 |
| 8 | 480 | 626 | 8 | 2 | 64 | 1 | 2 | 1.31e-3 | 160 | 6e18 |
| 8 | 512 | 512 | 8 | 2 | 64 | 1 | 2 | 1.31e-3 | 160 | 6e18 |
| 8 | 544 | 410 | 8 | 2 | 64 | 1 | 2 | 1.31e-3 | 160 | 6e18 |
| 8 | 560 | 364 | 8 | 2 | 64 | 1 | 2 | 1.31e-3 | 160 | 6e18 |
| 8 | 576 | 320 | 8 | 2 | 64 | 1 | 2 | 1.31e-3 | 160 | 6e18 |
| 10 | 600 | 766 | 10 | 2 | 64 | 1 | 2 | 1.11e-3 | 224 | 2e19 |
| 10 | 640 | 640 | 10 | 2 | 64 | 1 | 2 | 1.11e-3 | 224 | 2e19 |
| 10 | 680 | 528 | 10 | 2 | 64 | 1 | 2 | 1.11e-3 | 224 | 2e19 |
| 10 | 700 | 476 | 10 | 2 | 64 | 1 | 2 | 1.11e-3 | 224 | 2e19 |
| 10 | 740 | 380 | 10 | 2 | 64 | 1 | 2 | 1.11e-3 | 224 | 2e19 |
| 14 | 696 | 988 | 12 | 4 | 64 | 1 | 2 | 9.5e-3 | 320 | 6e19 |
| 14 | 768 | 768 | 12 | 4 | 64 | 1 | 2 | 9.5e-3 | 320 | 6e19 |
| 14 | 816 | 642 | 12 | 4 | 64 | 1 | 2 | 9.5e-3 | 320 | 6e19 |
| 14 | 840 | 584 | 12 | 4 | 64 | 1 | 2 | 9.5e-3 | 320 | 6e19 |
| 14 | 888 | 474 | 12 | 4 | 64 | 1 | 2 | 9.5e-3 | 320 | 6e19 |
| 16 | 896 | 1378 | 16 | 4 | 64 | 1 | 2 | 8.1e-3 | 512 | 2e20 |
| 16 | 1024 | 1024 | 16 | 4 | 64 | 1 | 2 | 8.1e-3 | 512 | 2e20 |
| 16 | 1088 | 876 | 16 | 4 | 64 | 1 | 2 | 8.1e-3 | 512 | 2e20 |
| 16 | 1152 | 742 | 16 | 4 | 64 | 1 | 2 | 8.1e-3 | 512 | 2e20 |
| 16 | 1184 | 680 | 16 | 4 | 64 | 1 | 2 | 8.1e-3 | 512 | 2e20 |
| 22 | 1120 | 1686 | 20 | 4 | 64 | 1 | 2 | 7.0e-3 | 768 | 6e20 |
| 22 | 1280 | 1280 | 20 | 4 | 64 | 1 | 2 | 7.0e-3 | 768 | 6e20 |
| 22 | 1360 | 1110 | 20 | 4 | 64 | 1 | 2 | 7.0e-3 | 768 | 6e20 |
| 22 | 1440 | 956 | 20 | 4 | 64 | 1 | 2 | 7.0e-3 | 768 | 6e20 |
| 22 | 1520 | 816 | 20 | 4 | 64 | 1 | 2 | 7.0e-3 | 768 | 6e20 |

Table 13: Representative parallelism configurations in our experiments.

| Model | $n_{\text{layers}}$ | $d_{\text{model}}$ | $d_{\text{ffn}}$ | $d_{\text{expert}}$ | $E$ | $E_a$ | **EP** | **TP** | **PP** |
|---|---|---|---|---|---|---|---|---|---|
| Experimental Model Example 1 | 8 | 384 | - | 96 | 256 | 8 | 8 | 1 | 1 |
| Experimental Model Example 2 | 8 | 512 | - | 128 | 256 | 8 | 8 | 1 | 1 |
| Experimental Model Example 3 | 10 | 640 | - | 160 | 256 | 8 | 8 | 1 | 1 |
| Experimental Model Example 4 | 14 | 768 | - | 192 | 256 | 8 | 8 | 1 | 1 |
| Experimental Model Example 5 | 16 | 1024 | - | 256 | 256 | 8 | 8 | 1 | 1 |
| Experimental Model Example 6 | 22 | 1280 | - | 320 | 256 | 8 | 8 | 1 | 2 |
| Dense-6.1B | 28 | 4096 | 14336 | - | - | - | - | 2 | 1 |
| MoE-mini | 20 | 2048 | 5120 | 384 | 384 | 12 | 8 | 2 | 1 |

Table 14: Experimental configurations for analyzing the impact of the number of attention heads ($n_{\text{head}}$). Within each experimental group, only $n_{\text{head}}$ is varied while other parameters remain fixed.

| $n_{layers}$ | $d_{model}$ | $d_{expert}$ | $n_{heads}$ | $n_{kv\_head}$ | $E$ | $E_a$ | $E_s$ | $\eta$ | $B$ | Max training FLOPs |
|---|---|---|---|---|---|---|---|---|---|---|
| 8 | 384 | 320 | [2,4,6,8,12] | 2 | 64 | 2 | 1 | 1.52e-3 | 98 | 2e18 |
| 8 | 512 | 512 | [2,4,8,12,16] | 2 | 64 | 2 | 1 | 1.31e-3 | 147 | 6e18 |
| 10 | 640 | 640 | [2,4,8,10,16,20] | 2 | 64 | 2 | 1 | 1.11e-3 | 228 | 2e19 |
| 14 | 768 | 768 | [6,8,12,16,24] | 4 | 64 | 2 | 1 | 9.5e-4 | 342 | 6e19 |
| 16 | 1024 | 1024 | [4,8,16,24.32] | 4 | 64 | 2 | 1 | 8.1e-4 | 531 | 2e20 |

Table 15: Impact of granularity on load balancing. Finer granularity (more experts) increases the coefficient of variation (CV), indicating greater load imbalance.

| $n_{layers}$ | $d_{model}$ | $n_{\text{heads}}$ | $E$ | $E_a$ | $E_s$ | $d_{\text{expert}}$ | $B$ | $\eta$ | Training FLOPs | CV |
|---|---|---|---|---|---|---|---|---|---|---|
| 14 | 768 | 12 | 192 | 6 | 3 | 384 | 342 | 9.5e-4 | 6e19 | 0.033 |
| 14 | 768 | 12 | 256 | 8 | 4 | 192 | 342 | 9.5e-4 | 6e19 | 0.052 |
| 14 | 768 | 12 | 384 | 12 | 6 | 128 | 342 | 9.5e-4 | 6e19 | 0.061 |
| 14 | 768 | 12 | 512 | 16 | 8 | 96 | 342 | 9.5e-4 | 6e19 | 0.093 |

Table 16: Goodness-of-fit comparison for the functional form relating EL and Activation Ratio ($A$). The saturating transformation provides a better fit.

| Functional Form | $R^2$ |
|---|---|
| Power-law with saturating transformation (**ours, Eq. 1**) | **0.9915** |
| Standard power-law (i.e., $\log \text{EL}(A) \propto \log A$) | 0.9772 |

Table 17: Goodness-of-fit comparison for the functional form relating Effective Loss (EL) and Expert Granularity ($G$). The log-polynomial function best captures the U-shaped trend.

| Functional Form | R-squared |
|---|---|
| Log-polynomial (**ours, Eq. 2**) | **0.9575** |
| Standard power-law (i.e., $\log \text{EL}(G) \propto \log G$) | 0.8276 |
| Power-law with saturating transformation | 0.9432 |

