# OpenReview forum: "Towards Greater Leverage: Scaling Laws for Efficient Mixture-of-Experts Language Models"
_ICLR.cc/2026/Conference — ICLR 2026 Poster_

### Official Review · Reviewer_WZ6e · 2025-10-21

**Soundness:** 2
**Presentation:** 3
**Contribution:** 3
**Rating:** 4
**Confidence:** 4

**Summary:**

The paper introduces Efficiency Leverage (EL), a metric designed to compare the training efficiency of sparse Mixture of Experts (MoE) versus dense Transformer models. The authors' primary contribution is developing a scaling law that expresses EL as a function of model architecture and compute budget and validating it on a moderately large computational scale.

The methodology proceeds in three phases:

1. **Establishing a testbed**: The authors develop predictors for optimal training hyperparameters, which allows them to reduce bias in the subsequent scaling law analysis.

2. **Optimal model/data allocation study**: They investigate how to optimally allocate model size and dataset size given a fixed compute budget, separately for MoE and dense architectures.

3. **Scaling law formulation**: Using 1. and 2., they derive and fit a joint scaling law across three variables: model sparsity, expert granularity, and total training budget (measured in FLOPs). They study marginal scaling behavior with respect to each variable individually before formulating the joint relationship.

The authors validate their scaling law in two ways. First, they present plots showing fit quality on a held-out validation set, though without goodness-of-fit metrics. Second, and more importantly, they conduct two large-scale training runs—one dense model and one sparse MoE model—trained on identical tokens using compute budgets significantly larger than those used to fit the scaling law. The validation succeeds: the models achieve comparable performance, yet the dense model requires more than 7× the active parameters, confirming the scaling law predictions (though I need explanation from the authors on the details).

Additionally, the paper analyzes how architectural choices affect scaling behavior, including the number of shared experts, expert granularity, and the proportion of attention versus MLP FLOPs.

**Strengths:**

**Conceptual contribution:**

EL has the potential to be a useful metric that simplifies efficiency comparisons between MoE and dense architectures. However, the metric needs to be precisely defined, and its usage throughout the paper must remain consistent with that definition, with any claims properly justified.

**Empirical validation:**

The validation experiments successfully corroborate the paper's claims. The authors train models at significantly larger scales than those used to fit the scaling law, and the predictions hold. This is a strong empirical result.

**Methodology:**

1. **Structured experimental foundation**: The authors take a systematic approach to building the foundation for EL scaling law experiments. They follow existing literature to develop two key components:
   - A system for choosing optimal hyperparameters with respect to compute budget. While they apply insights from only one MoE configuration to other architectures, they validate this transfer "in the vicinity" of 16× sparse MoE. The dense architecture is treated separately, so all comparisons are fair with regards to choice of core training hyperparameters.
   - Scaling laws for efficient budget allocation, once again done separately for dense and Moe models.

2. **Comprehensive architectural analysis**: Before deriving EL scaling laws, the authors systematically study key architectural factors including expert activation ratio, expert granularity, and shared expert ratio. This provides a solid foundation for understanding how these design choices affect the scaling behavior.

**Weaknesses:**

**Major Issue: Imprecise definition of EL and its inconsistent use throughout the paper, lack of clarity in experimental setup**

The definition of Efficiency Leverage (lines 186-188) lacks precision, and its usage throughout the paper creates confusion for readers. Several issues arise:

1. **Unclear functional form**: The formal definition of EL makes it a function of three arguments: MoE architecture, MoE compute budget, and dense architecture. However, the authors' usage throughout the text suggests a different understanding. Consider these statements:

   - Lines 16-17: "a metric quantifying the computational advantage of an MoE model over a dense equivalent"
   - Lines 25-27: "When trained on an identical 1T high-quality token dataset, MoE-mini matched the performance of the 6.1B dense model while consuming over 7× fewer computational resources, thereby confirming the accuracy of our scaling laws"
   - Lines 70-72: "an EL of 5, for example, means the MoE architecture performs like a dense model with five times the active parameters for a similar compute budget"

   These statements imply EL is an architecture-level property, independent of compute budget. Yet even under favorable assumptions (e.g., Chinchilla-like scaling), given the current definition, EL remains a function of compute budget and the chosen dense architecture.

2. **Missing justification**: Lines 185-189 make claims that do not clearly follow from the stated definition. To improve this, the authors should provide a complete derivation with explicit assumptions and literature references showing how EL reduces to the simpler form.

3. **Section 4 methodology unclear**:
   - How were the datapoints for EL obtained? What is the exact procedure for training dense models to reach the precise training loss of a given MoE run?
   - Figure 4 shows both fitted curves and datapoints, but their nature is ambiguous. Are these points from actual experiments or predictions? Figure 3 shows models up to ~3×10²⁰ FLOPs, yet Figure 4 plots extend to at least 1×10²² FLOPs (acknowledging that 4c is explicitly an extrapolation). If the points are predictions, this is misleading—the notation is not clearly specified, and the marked points suggest real experiments. If they are actual experiments, then the final validation run at 1×10²² FLOPs is not truly an extrapolation beyond the fitting range, which diminishes the claimed strength of the scaling law validation.

**Technical issues:**

- In the EL definition, the variable $C_{target}$ is never used and should be removed.
- The use of ε → 0 in the constraint is redundant. As ε approaches zero, this becomes an equality constraint. Why not write it as an equality directly?
- Sections 4.2.1 and 4.2.2. are concerned with fitting scaling laws for each variable separately, keeping the other two fixed, and fitting the joint form, respectively. Thus, the name of 4.2.1 should rather be called either marginal or univariate scaling laws. Separable means its already a joint function (so, of all variables), but it's of a special form functional form.
**Unsupported claim:**

>Lines [249-251]: "It represents a critical balance: [...] while overly coarse-grained ones fail to achieve effective specialization."

This claim about expert granularity is stated as fact but lacks supporting analysis or evidence in the paper. The only thing that is supported is that increased granularity deteriorates model quality from some point on - though, judging by the analysis in Figure 13, it is possible that simply having the balancing loss be a function of granularity (more experts -> more balancing needed) will make granularity once again effective.

**Questions:**

# Questions for Authors

1. **What is the precise definition of EL?** Can you provide a complete mathematical definition that reconciles the formal definition in lines 186-188 with the simpler characterizations used throughout the paper (e.g., "computational advantage" or "performs like a dense model with N times the active parameters")? Under what assumptions does EL become independent of compute budget?

2. **How did you generate the (MoE, dense) run pairs for EL datapoints?** What is the exact methodology for pairing runs? Do you train dense models specifically to match the loss of each MoE run, or do you use some form of interpolation/prediction?

3. **Do you drop tokens at any point?** Please clarify whether token dropping occurs and if any expert capacity mechanism is implemented during training, evaluation, or both.

4. **What is the quality of fit for your scaling laws?** The plots show that test performance appears worse than train performance, but there is no discussion of fit quality. Did you perform a detailed analysis similar to, for example, Besiroglu et al. (2024)? Specifically:
   - What are the RMSE values for train and test?
   - How does fit quality vary across different dimensions (model size, granularity, sparsity)?
   - What do the marginal distributions look like?
   - Are there systematic patterns in the residuals?
5. **How was load balancing implemented?** Was it run locally, or were the routing statistics all-reduced across e.g. the expert parallel group or globally? Please provide implementation details.
6. **What is your intuition behind granularity performing worse beyond a certain point?** The paper states that "excessively fine-grained experts suffer from insufficient capacity, while overly coarse-grained ones fail to achieve effective specialization." What evidence or analysis supports this claim? I was only able to find an anlysis on model quality (training perplexity), but without any analysis on specialization or otherwise.
---

## References

Besiroglu, T., Erdil, E., Barnett, M., & You, J. (2024). Chinchilla Scaling: A replication attempt. arXiv preprint arXiv:2404.10102.

---

> ### Author Response · Authors · 2025-11-22
> **Response to Weaknesses (1/3)**
>
> > **Weakness 1: Unclear functional form of EL. The formal definition of EL makes it a function of three arguments: MoE architecture, MoE compute budget, and dense architecture. However, the authors' usage throughout the text suggests a different understanding.**
>
> **Response:**  We sincerely thank you for this insightful comment. The definition of EL (Definition 3.1 in the initial manuscript) is a function of the MoE architecture, the dense architecture, and the compute budget. It is not an inherently compute-independent.
>
> The apparent discrepancy you noted stems from a empirical finding: under training conditions where **MoE and dense models are trained on the same token budget and achieve comparable loss**, the EL value approximates the ratio of the dense model’s parameters to the MoE’s active parameters. Specifically, since compute budget $C \propto N_a \times D$ (active parameters × data tokens) and when $D_{\text{MoE}} = D_{\text{dense}}$ (as in our 1T-token validation experiments in Section 5), the compute ratio simplifies to $C_{\text{dense}} / C_{\text{MoE}} \approx N_{\text{dense}} / N_{\text{MoE}(\text{active})}$. Thus, $\mathrm{EL} \approx N_{\text{dense}} / N_{\text{MoE}(\text{active})}$ in such aligned settings, enabling the intuitive interpretation as a "parameter efficiency multiplier."
>
> Therefore, statements such as "EL quantifies computational advantage" were intended as **practical, scenario-specific interpretations based on this empirical regularity, not as revisions to the formal definition**. We recognize that this distinction was not made sufficiently clear. We will revise the manuscript to eliminate any ambiguity.
>
> &nbsp;
>
> > **Weakness 2: Missing justification: Lines 185-189 make claims that do not clearly follow from the stated definition.**
>
> **Response:** Thank  you for this valuable suggestion. We provide the requested derivation below and clarify the underlying assumptions.
>
> 1. **Formal Definition and Justification.**
>
> Let $L_{\mathcal{X}}(C)$ denote the best achievable evaluation loss for architecture $\mathcal{X}$ trained with compute budget $C$, assuming optimally tuned hyperparameters. In practice, $L_{\mathcal{X}}(C)$ is often  modeled by a power law, e.g., $L_{\mathcal{X}}(C) = \alpha_{\mathcal{X}}C^{\beta_{\mathcal{X}}} +  b_{\mathcal{X}}$, as previous works [1,2,3]. For a target loss $L^\star$ in a regime where $L_{\mathcal{X}}(C)$ is continuous and strictly monotonically decreasing, define the minimal compute required to attain $L^\star$ as the inverse function:
>
> \begin{equation\*}
>          C_{\mathcal{X}}(L^\star) \equiv L_{\mathcal{X}}^{-1}(L^\star)
> \end{equation\*}
>
> The EL of an MoE architecture $\mathcal{X}\_{MoE}$ relative to a dense architecture ${\mathcal{X}}\_{Dense}$, evaluated at a target loss $L^\star$, is the ratio of their required compute budgets:
>
> \begin{equation\*}
> \mathrm{EL}(\mathcal{X}\_{\mathrm{MoE}} \mid \mathcal{X}\_{\mathrm{Dense}}; L^\star)=\frac{C\_{\mathcal{X}\_{\mathrm{Dense}}}(L^\star)}{C\_{\mathcal{X}\_{\mathrm{MoE}}}(L^\star)}.
> \end{equation\*}
>
> **Setting the target loss to the loss achieved by the MoE model at compute $C\_{\mathrm{MoE}}$, i.e., $L^\star = L\_{\mathcal{X}\_{\mathrm{MoE}}}(C_{\mathrm{MoE}})$**, yields:
>
> \begin{equation\*}
> \mathrm{EL}(\mathcal{X}\_{\mathrm{MoE}} \mid \mathcal{X}\_{\mathrm{Dense}}; C\_{\mathrm{MoE}}) = \frac{L\_{\mathcal{X}\_{\mathrm{Dense}}}^{-1}\left(L\_{\mathcal{X}\_{\mathrm{MoE}}}(C\_{\mathrm{MoE}})\right)}{C\_{\mathrm{MoE}}}  = \frac{C\_{\mathrm{Dense}}}{C\_{\mathrm{MoE}}},
> \end{equation\*}
>
> where $C\_{\mathrm{Dense}} = L\_{\mathcal{X}\_{\mathrm{Dense}}}^{-1}\left(L\_{\mathcal{X}\_{\mathrm{MoE}}}(C\_{\mathrm{MoE}})\right)$ is the compute budget required by the dense model to match the MoE loss at budget $C\_{\mathrm{MoE}}$. This reduction follows directly from the definition of $C\_{\mathcal{X}}(L^\star)$ and the existence of $L\_{\mathcal{X}}^{-1}$ under continuity and strict monotonicity.
>
> 2. **Assumptions and Empirical Estimation.**
>
> The following assumptions underpin our framework and its empirical application:
>
>    *   **Compute Measurement:** Compute budget ($C$) is measured in total theoretical training FLOPs, consistently across all models. This excludes costs such as interconnect overhead, memory traffic, and other wall-clock factors not captured by arithmetic FLOPs.
>
>    * **Empirical Modeling:** To estimate $L_{\mathcal{X}}(C)$ from a finite set of runs, we fit an architecture-specific loss scaling law (e.g., $L_{\mathcal{X}}(C) = \alpha_{\mathcal{X}} C^{\beta_{\mathcal{X}}}+  b_{\mathcal{X}}$) for a given configuration, as previous works [1,2,3]. Crucially, the theoretical definition of EL is form-agnostic, but its empirical estimation requires this function-fitting step.
>
> We will revise the manuscript accordingly, stating the above assumptions explicitly, and adding the cited literature make our methodology clearer and more justifiable.

---

> ### Author Response · Authors · 2025-11-22
> **Response to Weaknesses (2/3)**
>
> > **Weakness 3.1: How were the datapoints for EL obtained? What is the exact procedure for training dense models to reach the precise training loss of a given MoE run?**
>
> **Response:** As clarified in our responses to Weakness 1 and Weakness 2, our methodology does not involve training a unique dense model to match the loss of each individual MoE run. Instead, we employ a more systematic and scalable approach based on modeling the loss-compute scaling behavior for each model family. For a specific MoE architecture (e.g., for a fixed activation ratio and granularity), the process of obtaining data points for EL is as follows:
>
> 1. **Collecting `(compute budget, optimal loss)` Data:**
>
>    We first train a suite of MoE models (see Tables 8–11) and a corresponding suite of dense counterparts. For a given MoE configuration, its dense counterpart is defined as a standard Transformer architecture, equivalent to an MoE model with a 100% activation rate. **As shown in Table 8, the model architecture with an activation rate of 1.0 serves as the dense counterpart for all other MoE configurations in our study.** All models are trained on the same dataset with the same recipe, for up to $3\times 10^{20}$ FLOPs. This process generates a set of '(compute, optimal loss)' data points $\{(C, \ell)\}$ for both the specific MoE architecture $\mathcal{X}\_{\mathrm{MoE}}$ and dense architecture $\mathcal{X}\_{\mathrm{Dense}}$. This is illustrated in Figure 11(a), Figure 12(a), and Figure 14(a).
>
> 2. **Fitting Loss Scaling Curves:**
>
>    We then fit separate loss scaling functions, $L\_{\mathcal{X}\_{\mathrm{MoE}}}(\cdot)$ and $L\_{\mathcal{X}\_{\mathrm{Dense}}}(\cdot)$, to the collected data for specific MoE  and dense architecture, respectively. We use a standard power-law form,  $L\_{\mathcal{X}}(C) = \alpha\_{\mathcal{X}}C^{\beta\_{\mathcal{X}}} +  b\_{\mathcal{X}}$, consistent with prior work [1,2,3]. This process yields smooth loss scaling functions that can predict the architecture's optimal loss at any given compute budget $C$. such as illustrate by Figure 11(a), Figure 12(a) and Figure 14(a). These fitted curves are shown in the left panels of Figure 11(b), Figure 12(b), and Figure 14(b).
>
> 3. **Computing EL by Loss Matching:**
>
>    For an MoE at $C\_{\text{MoE}}$, we first use its fitted curve to calculate the predicted loss: $\mathcal{L}^\star=L\_{\mathcal{X}\_{\mathrm{MoE}}}(C\_{\text{MoE}})$. Next, we use the dense model's loss curve, $L\_{\mathcal{X}\_{\mathrm{Dense}}}(C)$, to find the compute required to achieve the same loss by solving $C\_{\mathrm{Dense}} = L\_{\mathcal{X}\_{\mathrm{Dense}}}^{-1}\left(\mathcal{L}^\star\right)$. Finally, we compute the EL as defined: $\mathrm{EL} = C\_{\text{dense}}/C\_{\text{MoE}}$. These EL are shown in the right panels of Figure 11(b), Figure 12(b), and Figure 14(b).
>
> This methodology allows us to systematically and reproducibly evaluate EL across a continuous range of compute budgets. The results of each step are provided in **Appendix F of the initial manuscript**. We will move this detailed procedure to the main body of the paper and provide **pseudocode in the Appendix K** to ensure full transparency.
>
> &nbsp;
>
> > **Weakness 3.2: Figure 4 shows both fitted curves and datapoints, but their nature is ambiguous. Are these points from actual experiments or predictions?**
>
> **Response:** We appreciate the opportunity to clarify Figure 4.
>
> * **Data vs. predictions:** In Figure 4, filled markers up to $3\times 10^{20}$ FLOPs denote experimental datapoints used for fitting. Markers beyond $3\times 10^{20}$ FLOPs are extrapolated predictions obtained by evaluating $L\_{\text{MoE}}(C)$ and $L\_{\text{dense}}(C)$ and computing EL via inversion as described above.
>
> * **Strength of validation:** Our final large-scale validation run at $1\times 10^{22}$ FLOPs is a real, held-out experiment conducted after fitting with data only up to $3\times 10^{20}$ FLOPs. It serves as a **true out-of-domain test (over 30$\times$ in compute)** of the fitted curves’ predictive validity.
>
> We will revise the caption of Figure 4 to explicitly state which points are experimental versus predicted.

---

> ### Author Response · Authors · 2025-11-22
> **Response to Weaknesses (3/3)**
>
> > **Weakness 4: Technical issues**
>
> **Response:** Thank you for the clear technical feedback. We will implement the following revisions:
>
> * **Definition and constraints in Eq 2:** We will fix the typos and corrected the notation from $C\_{target}$ to $C\_{\text{MoE}}$ for consistency. The earlier use of $\varepsilon \to 0$ was meant to capture numerical tolerance and measurement noise; we will revise Definition 3.1 to aligns with our derivation in Weakness 2.
>
> * **Section naming and use of "separable":** Our initial choice of "Separable Scaling Laws" was intended to maintain consistency with prior work [4]. Upon reflection, we agree with the reviewer and retitled Section 4.2.1 to “Univariate Scaling Laws.”
>
> We are grateful for these detailed comments. **These edits tighten the formal presentation and terminology without affecting any empirical outcomes or conclusions**.
>
> &nbsp;
>
> > **Weakness 5: Unsupported claim: The claim in Lines [249-251] about expert granularity is stated as fact but lacks supporting analysis or evidence in the paper.**
>
> **Response:** Thank you for this insightful point. The balancing loss coefficient was indeed held constant in our main analysis, precisely to isolate and study the effect of granularity in a controlled setting. Your hypothesis—that performance could be restored by making this coefficient a function of granularity—is highly plausible.
>
> * **Regarding "Excessively Fine-Grained"**: When we employed an weaker routing mechanism, we observed that the optimal granularity shifted towards coarser partitions (as seen by comparing Figure 12 and Figure 13). We observed that increasing expert granularity (i.e., using more, smaller experts) leads to a less uniform token distribution, measured by a higher Coefficient of Variation ($CV$) of the expert loads, where $CV = \frac{\text{standard deviation of the token distribution}}{\text{mean of the token distribution}}$. The following table provides empirical evidence for this.
>    | model | $n\_{layers}$ | $d\_{model}$ | $E$ | $E\_a$ | $d\_{expert}$ | CV (coefficient of variation) |
>    |:---|---:|---:|---:|---:|---:|---:|
>    | Model Example 1 | 14 | 768 | 128 | 4 | 384 | 0.033 |
>    | Model Example 2 | 14 | 768 | 256 | 8 | 192 | 0.052 |
>    | Model Example 3 | 14 | 768 | 384 | 12 | 128 | 0.061 |
>    | Model Example 4 | 14 | 768 | 512 | 16 | 96 | 0.093 |
>
> * **Regarding "Overly Coarse-Grained"**: Our statement about “overly coarse” experts draws on prior work [5], which empirically shows that finer-grained partitioning enhances expert specialization. We will clarify this connection and distinguish specialization benefits from utilization/balancing challenges.
>
> We will revise the paper to clarify these points and will rephrase our claim to be more precise.
>
> &nbsp;
>
> ---
>
> **References:**
>
> [1] Kaplan, Jared, et al. "Scaling laws for neural language models." arXiv preprint arXiv:2001.08361 (2020).
>
> [2] Henighan, Tom, et al. "Scaling laws for autoregressive generative modeling." arXiv preprint arXiv:2010.14701 (2020).
>
> [3] OpenAI, et al. "Gpt-4 technical report." arXiv preprint arXiv:2303.08774 (2023).
>
> [4] Clark, Aidan, et al. "Unified scaling laws for routed language models." ICML. PMLR, 2022.
>
> [5] Dai, Damai, et al. "Deepseekmoe: Towards ultimate expert specialization in mixture-of-experts language models." arXiv preprint arXiv:2401.06066 (2024).

---

> ### Author Response · Authors · 2025-11-22
> **Response to Questions (1/2)**
>
> > **Question 1: What is the precise definition of EL? Under what assumptions does EL become independent of compute budget?**
>
> **Response:** Thank you for this crucial follow-up question. Summarizing our detailed responses to Weakness 1 and Weakness 2, we offer the following clarification.
>
> * **Definition of EL.**
>
> Formally, the EL is defined as the ratio of the compute budgets ($C$) required for a dense and an MoE architecture to achieve the exact same target loss value, $L^\star$. Let $L\_{\mathcal{X}}(C)$ be the optimal loss scaling function for an architecture $\mathcal{X}$, representing the best possible loss for a given compute budget $C$. This function is not from a single run but is empirically modeled from the performance frontier, consistent with scaling literature [1,2,3]. The compute needed to reach loss $L^\star$ is the inverse, i.e., $C\_{\mathcal{X}}(L^\star) = L\_{\mathcal{X}}^{-1}(L^\star)$. The EL is then:
>
> \begin{equation*}
> \mathrm{EL}(\mathcal{X}\_{\mathrm{MoE}} \mid \mathcal{X}\_{\mathrm{Dense}}; L^\star) = \frac{C\_{\mathcal{X}\_{\mathrm{Dense}}}(L^\star)}{C\_{\mathcal{X}\_{\mathrm{MoE}}}(L^\star)}
> \end{equation*}
>
> In our work, we set $L^\star$ to the loss achieved by the MoE with its budget $C\_{\mathrm{MoE}}$ (i.e., $L^\star = L\_{\mathcal{X}\_{\mathrm{MoE}}}(C\_{\mathrm{MoE}})$), which simplifies the expression to:
>
> \begin{equation*}
> \mathrm{EL}(\mathcal{X}\_{\mathrm{MoE}} \mid \mathcal{X}\_{\mathrm{Dense}}; C\_{\mathrm{MoE}})=\frac{L\_{\mathcal{X}\_{\mathrm{Dense}}}^{-1}\left(L\_{\mathcal{X}\_{\mathrm{MoE}}}(C\_{\mathrm{MoE}})\right)}{C\_{\mathrm{MoE}}}
> =\frac{C\_{\mathrm{Dense}}}{C\_{\mathrm{MoE}}},
> \end{equation*}
>
> where $C\_{\mathrm{Dense}}$ is the compute the dense counterpart requires to match the MoE's loss. By this formal definition, EL is fundamentally a function of the architectures and the compute budget.
>
> * **When EL becomes (approximately) compute-independent.**
>
> EL is not inherently compute-independent. It approximates a constant value only under a specific, practical assumption: that the MoE and dense models are trained on the same number of tokens $D\_{\text{MoE}} = D\_{\text{dense}}$. This interpretation is based on an empirical regularity, not a change to the formal definition. Because the compute budget $C \propto N\_{active} \times D$ (active parameters × data tokens), matching the token budget ($D$) means the compute ratio simplifies to the ratio of parameters: $\mathrm{EL} = C\_{\text{dense}} / C\_{\text{MoE}} \approx N\_{\text{dense}} / N_{\text{MoE}(\text{active})}$. This allows the intuitive interpretation as a "parameter efficiency multiplier" in this specific, practical scenario.
>
> We will revise the manuscript for greater precision.
>
> &nbsp;
>
> > **Question 2: How did you generate the (MoE, dense) run pairs for EL datapoints? What is the exact methodology for pairing runs? Do you train dense models specifically to match the loss of each MoE run, or do you use some form of interpolation/prediction?**
>
> **Response:** As we detail in our full response to Weakness 3.1, we do not train dense models specifically to match the loss of each MoE run. Instead, our method for generating (MoE, dense) run pairs relies on interpolation via fitted loss scaling laws. In essence, the key steps are:
>
> 1. **Data Collection:** We first independently train two separate families of models—a suite of MoE models and a suite of dense counterparts—to collect two distinct sets of '(compute, loss)' data points, as illustrated in Figures 11(a), 12(a), and 14(a).
>
> 2. **Fitting Loss Scaling Curves:** We then fit a continuous loss scaling curve to each family's data, yielding two predictive functions: $L\_{\mathcal{X}\_{\mathrm{MoE}}}(C)$ and $L\_{\mathcal{X}\_{\mathrm{Dense}}}(C)$, as illustrated in the left panels of Figures 11(b), 12(b), and 14(b).
>
> 3. **Computing EL by Loss Matching:** Given a compute $C_{\text{MoE}}$, we compute its target loss $L^* = L\_{\mathcal{X}\_{\mathrm{MoE}}}(C\_{\text{MoE}})$, then invert the dense curve to obtain $C\_{\text{dense}}$ satisfying $L\_{\mathcal{X}\_{\mathrm{Dense}}}(C\_{\text{dense}}) = L^*$. The EL datapoint is $\mathrm{EL} = C\_{\text{dense}} / C\_{\text{MoE}}$, as illustrated in the right panels of Figures 11(b), 12(b), and 14(b).
>
> This approach allows us to evaluate EL across a continuous range without needing to perform expensive one-to-one training for each data point. We will clarify this process in the revised manuscript.
>
> &nbsp;
>
> > **Question 3: Do you drop tokens at any point? Please clarify whether token dropping occurs and if any expert capacity mechanism is implemented during training, evaluation, or both.**
>
> **Response:** We do not employ token dropping or expert capacity mechanisms at any stage of our experiments.

---

> ### Author Response · Authors · 2025-11-22
> **Response to Questions (2/2)**
>
> > **Question 4: What is the quality of fit for your scaling laws? The plots show that test performance appears worse than train performance, but there is no discussion of fit quality. Did you perform a detailed analysis similar to, for example, Besiroglu et al. (2024)?**
>
> **Response:**  Yes. We did perform a detailed analysis on the quality of fit, and the results confirm a strong model performance. We will incorporate these findings into the revised manuscript. Key metrics include:
>
> * **RMSE:** The training set RMSE is 0.2169 (over 200 point), while the validation set RMSE is 0.5275 (24 points).
>
> * **Goodness-of-Fit:** Our model achieves a high overall R-Squared of 0.9858. The R-Squared values for the sparsity and granularity dimensions are 0.9915 and 0.9575, respectively.
>
> * **Marginal Distributions and Systematic Patterns in the Residuals:** The residuals are approximately normally distributed and centered at zero (mean: -0.0273, std: 0.2803). A plot of residuals against predicted values shows no discernible patterns. The plot of residuals will be added to the appendix in the revision.
>
> &nbsp;
>
>
> > **Question 5: How was load balancing implemented? Was it run locally, or were the routing statistics all-reduced across e.g. the expert parallel group or globally? Please provide implementation details.**
>
> **Response:** Our load balancing implementation follows the standard practice of open-source frameworks like Megatron-LM[6] and Megablocks[7], calculating the balancing loss at the **micro-batch level**. We are also actively exploring a global-batch level calculation [8]. Based on our prior observations (see response to W5 and Appendix F.2), we hypothesize that **more advanced routing algorithms will favor an even finer expert granularity**.
>
> As different architectures yield distinct loss-scaling curves, changes in  implementation will inevitably affect the EL distribution. We will emphasize these details and aim to provide corresponding results based on the latest architectural changes in a future update.
>
> &nbsp;
>
> > **Question 6: What is your intuition behind granularity performing worse beyond a certain point? The paper states that "excessively fine-grained experts suffer from insufficient capacity, while overly coarse-grained ones fail to achieve effective specialization." What evidence or analysis supports this claim? I was only able to find an anlysis on model quality (training perplexity), but without any analysis on specialization or otherwise.**
>
> **Response:** Our response to Weakness 5 clarify this point. Generally, our intuition is a trade-off between specialization and load balancing, supported by:
>
> 1) **Regarding "Overly Coarse-Grained"**: Our claim here builds on established findings from prior work [5], which empirically demonstrated that coarse partitioning limits effective expert specialization.
>
> 2) **Regarding "Excessively Fine-Grained"**: Our analysis shows that finer granularity increases load imbalance (higher CV of expert loads). This leads to under-trained experts and degrades overall model quality in our experiments.
>
> Thus, our analysis identifies the performance limit due to load balancing, while prior work [5] identifies the limit due to poor specialization. We will revise the paper to clarify these points.
>
> &nbsp;
>
> ---
>
> **References:**
>
> [1] Kaplan, Jared, et al. "Scaling laws for neural language models." arXiv preprint arXiv:2001.08361 (2020).
>
> [2] Henighan, Tom, et al. "Scaling laws for autoregressive generative modeling." arXiv preprint arXiv:2010.14701 (2020).
>
> [3] OpenAI, et al. "Gpt-4 technical report." arXiv preprint arXiv:2303.08774 (2023).
>
> [4] Clark, Aidan, et al. "Unified scaling laws for routed language models." ICML. PMLR, 2022.
>
> [5] Dai, Damai, et al. "Deepseekmoe: Towards ultimate expert specialization in mixture-of-experts language models." arXiv preprint arXiv:2401.06066 (2024).
>
> [6] Shoeybi, Mohammad, et al. "Megatron-lm: Training multi-billion parameter language models using model parallelism." arXiv preprint arXiv:1909.08053 (2019).
>
> [7] Gale, Trevor, et al. "Megablocks: Efficient sparse training with mixture-of-experts." Proceedings of Machine Learning and Systems 5 (2023): 288-304.
>
> [8] Qiu, Zihan, et al. "Demons in the detail: On implementing load balancing loss for training specialized mixture-of-expert models." ACL 2025.

---

### Official Review · Reviewer_7T91 · 2025-10-30

**Soundness:** 3
**Presentation:** 3
**Contribution:** 3
**Rating:** 6
**Confidence:** 4

**Summary:**

This paper introduces 'Efficiency Leverage' (EL) to quantify the computational advantage of MoE models over dense models . It comprehensively investigates the scaling laws for MoE models, analyzing key factors such as expert granularity and activation ratio (sparsity). The study is supported by extensive data and figures and validates its findings by training a high-performing MoE model (MoE-mini) that confirms the predicted efficiency gains. The work provides a practical guide for MoE model design.

**Strengths:**

1. This work's originality stems from its formulation of "Efficiency Leverage" (EL), a clear metric to quantify MoE computational advantage . It uses this concept to build a unified scaling law for EL, connecting it to the compute budget, activation ratio, and granularity.

2. The empirical quality is high, supported by over 300 trained models. A key strength is the preliminary work in Section 2, which derives MoE-specific scaling laws for optimal hyperparameters and data allocation . This step ensures a fair comparison, addressing a methodological flaw in prior studies.

3. The paper is clearly structured, and its claims are well-supported by data. The final validation, MoE-mini, successfully confirmed the law's >7x efficiency prediction . This demonstrates the work's significance as a practical guide for designing future efficient MoE models.

**Weaknesses:**

1. The conclusion that efficiency monotonically increases with sparsity (Key Takeaway 1) is based on a theoretical FLOPs model. This omits the practical wall-clock costs of routing, communication (e.g., all-to-all), and memory bandwidth for loading many distinct expert weights, which can become bottlenecks at high sparsity.

2. The functional forms for the scaling laws (Eq. 2, 3, 4, and 5) are presented without strong justification. It is not clear why these specific complex forms (e.g., log-polynomial for $G$, saturating transform for $A$) were chosen over simpler alternatives, or how the goodness-of-fit was evaluated.

**Questions:**

1. The derivation of optimal hyperparameters in Section 2.2 scales them as a function of total compute $C$, adapting a methodology from dense models. However, the paper notes that MoE backpropagation is different, as experts see only a subset of tokens . Why should the optimal batch size and learning rate scale only with total $C$, rather than also depending on MoE-specific factors (like activation ratio) that directly influence gradient statistics for each expert?

2. Could the authors provide more details on the fitting process for the scaling laws (Eq. 2, 3, 5)? Specifically, what was the rationale for choosing the log-polynomial form for granularity ? Were simpler functional forms tested, and how did their fit quality compare?

3. Regarding the finding that EL improves as activation ratio decreases (Key Takeaway 1) : Do the authors expect this trend to break down at a certain point when considering practical implementation overheads (e.g., wall-clock time) instead of just theoretical FLOPs?

---

> ### Author Response · Authors · 2025-11-22
> **Response to Weaknesses**
>
> > **Weakness 1: The conclusion that efficiency monotonically increases with sparsity omits the practical wall-clock costs.**
>
> **Response:**  We thank the reviewer for this crucial point, which clearly stems from valuable real-world engineering experience. **Our work deliberately abstracts away from these system-specific, engineering complexities.** A full end-to-end efficiency analysis would be **intractable**, as it depends heavily on specific hardware, network topology, and parallelization strategies. Therefore, our primary goal was to isolate and model the theoretical architectural advantage of MoE from a pure computational (FLOPs) perspective. This controlled approach allows us to establish a clear, theoretical upper bound on the benefits of sparsity, which is **a necessary first step before considering implementation-specific costs.**
>
> In essence, our model defines the **algorithmic potential**, while the engineering factors you mention determine the **realized performance**. The optimal architectural choice for any given application is therefore a **trade-off between maximizing this theoretical potential and accommodating practical system constraints**. We believe our contribution is a critical foundation toward predicting end-to-end training efficiency. We acknowledged this limitation **in the "Limitations" section of our initial manuscript** and will enhance this discussion in the revision.
>
> &nbsp;
>
> > **Weakness 2: The functional forms for the scaling laws are presented without strong justification.**
>
> **Response:** Thank you for this crucial question. **Our methodology, consistent with established scaling law research [1,2,3,4,5], is empirical.** For each law, we first identified the underlying trend from data visualizations and then rigorously validated our choice of function by comparing its goodness-of-fit against simpler alternatives. Here is a breakdown of our reasoning for each form:
>
> 1. **Saturating Transform for Activation Ratio ($A$) in Eq. 2:**
> We empirically observed that the benefits of decreasing the activation ratio show **diminishing returns**, especially at very low ratios. A saturating transformation, also employed by Clark et al. [1] to model similar phenomena, captures this effect perfectly. To validate this, we compared it against simpler models (e.g., standard power-law), as confirmed by the R-Squared values below, where the saturating form achieved a significantly better fit.
>
>     | Functional Form between EL and Activation Ratio ($A$). | R-Squared |
>     |:---|---:|
>     | power-law with saturating transformation (**ours, Eq.2 in the paper**) | 0.9915 |
>     | standard power-law (i.e., $\log{EL}_{C,G}( A )  = a_{A} \log  A$) | 0.9772 |
>
> 2. **Log-Polynomial for Granularity ($G$) in Eq. 3:**
> Our data revealed a **U-shaped relationship** between granularity and performance, where EL first increases and then decreases. A quadratic log-polynomial function is a standard and parsimonious choice for modeling such non-monotonic trends and is widely used in prior work [2,3]. We validated this by comparing its fit against other typical functions and the results below show that the log-polynomial form provides the most accurate fit.
>
>     | Functional Form between EL and  Expert Granularity ($G$). | R-Squared |
>     |:---|---:|
>     | log-polynomial (**ours, Eq.3 in the paper**) | 0.9575 |
>     | standard power-law (i.e., $\log{EL}_{C,A}( G )  = a_{G} \log  G$) | 0.8276 |
>     | power-law with saturating transformation (i.e., $\log{EL}_{C,A}( \hat G )  = a_{\hat G} \log  \hat  G$) | 0.9432 |
>
>
> 3. **Power Law for Compute Budget ($C$) in Eq. 4:**
> This choice was directly guided by our data. As shown in Figure 4c, the log-log plot of Effective Loss (EL) versus FLOPs exhibits a clear linear relationship, which is the definitive characteristic of a power-law dependency.
>
> 4. **Joint Scaling Law in Eq. 5:**
> The final law is an **empirical composite model designed to synthesize our individual findings**. It is not an arbitrary construction but reflects the observed interactions: The term for Activation Ratio ($A$) serves as the primary driver of efficiency. This is then modulated by the independent, non-linear adjustment from Granularity ($G$). Finally, the entire efficiency gain is amplified by the Compute Budget ($C$) through the overarching power-law pattern. This composite structure provides the most comprehensive explanation of the joint effects. Its high accuracy is empirically validated in **Figure 5**, achieving an R-squared of 0.9858. The model demonstrates strong predictive power with a low RMSE on both the training set (0.2169 over 200 points) and the validation set (0.5275 over 24 points).
>
> To address your concern directly, we will clarify the goodness-of-fit in the the revised manuscript. We appreciate your feedback in helping us improve the paper's rigor.

---

> ### Author Response · Authors · 2025-11-22
> **Response to Questions**
>
> > **Question 1: Why should the optimal batch size and learning rate scale only with total C, rather than also depending on MoE-specific factors (like activation ratio) that directly influence gradient statistics for each expert?**
>
> **Response:** Thank you for this insightful question. We agree that, ideally, optimal hyperparameters should account for MoE-specific factors like the activation ratio. However, **a full-scale search for every configuration is computationally unaffordable**. Our work addresses this challenge with a pragmatic and principled two-step methodology, **detailed in Section 2.2 and Appendix E.1**:
> 1. **Establish Foundational Scaling Laws:**  We first derive optimal scaling laws for batch size and learning rate based on total compute for a representative activation ratio.
> 2. **Verify Generalization across Other Activation Ratio:** We then empirically demonstrate that these laws generalize effectively across a range of other activation ratios with minimal performance degradation, as shown in **Figure 9**.
>
> This methodology is a significant improvement over prior MoE scaling law studies [2,3] that often uses uniform learning rate or batch size, and it ensures a fairer comparison by evaluating each model near its peak performance.
>
> To provide context on the cost, our hyperparameter search and validation experiments **already consumed over 300k GPU hours**. A more comprehensive study to establish fine-grained scaling laws for each distinct sparsity level would be prohibitively expensive, with an estimated **cost exceeding 1 million GPU hours** [4].
>
> We acknowledge this limitation and identify it as a valuable direction for future research in the revised manuscript.
>
> &nbsp;
>
> > **Question 2: Could the authors provide more details on the fitting process for the scaling laws, Specifically for for choosing the log-polynomial form for granularity ?**
>
> **Response:** Our response to Weakness 2 outlines the detailed rationale. Our methodology is empirical and follows standard practice in scaling law research [1,2,3,4,5]. It involves a two-step process for each law: 1) identifying the underlying trend from data visualizations, and 2) rigorously comparing the chosen function's goodness-of-fit against simpler alternatives.
>
> 1. **Rationale for the Log-Polynomial Form for Granularity (Eq. 3):** The choice of a quadratic log-polynomial was directly driven by our empirical observations. Our data revealed a clear U-shaped relationship between granularity and performance, where the effective loss first improves (decreases) and then worsens (increases) as granularity grows. A quadratic log-polynomial is a standard and parsimonious choice for modeling such non-monotonic phenomena and has been successfully used in prior work [2,3].
>
> 2. **Comparison with Simpler Functional Forms:** Yes, we explicitly tested against simpler functional forms. As stated in our response to Weakness 2, we validated our choice by comparing its fit against other typical functions. Our results showed that the log-polynomial form provided the most accurate fit for the observed data.
>
> To provide full transparency, we will clarify this entire process and the goodness-of-fit in the revised manuscript.
>
> &nbsp;
>
> **References:**
>
> [1] Clark, Aidan, et al. "Unified scaling laws for routed language models." International conference on machine learning. PMLR, 2022.
>
> [2] Krajewski, Jakub, et al. "Scaling laws for fine-grained mixture of experts." arXiv preprint arXiv:2402.07871 (2024).
>
> [3] Abnar, Samira, et al. "Parameters vs flops: Scaling laws for optimal sparsity for mixture-of-experts language models." arXiv preprint arXiv:2501.12370 (2025).
>
> [4] Li, Houyi, et al. "Predictable Scale: Part I--Optimal Hyperparameter Scaling Law in Large Language Model Pretraining." arXiv e-prints (2025): arXiv-2503.
>
> [5] Hoffmann, Jordan, et al. "Training compute-optimal large language models." arXiv preprint arXiv:2203.15556 (2022).

---

### Official Review · Reviewer_k9rc · 2025-10-31

**Soundness:** 3
**Presentation:** 4
**Contribution:** 3
**Rating:** 6
**Confidence:** 4

**Summary:**

This paper studies the effectiveness of training MoE versus dense models from the compute perspective.

First, a detailed ablation study is performed to search for the best architecture for training an MoE, considering shared experts, granularity, sparsity.
Then, scaling laws of the leverage (gain of performance wrt dense model) are established by considering the above factors independently.
Finally, the model is trained at the scale of around 20B total-1B active MoE to validate the performances.

**Strengths:**

- The study is of large scale (20B moe and 1T tokens) and detailed ablation studies justify their choices of architecture.
- A principled approach (scaling law) is used to tune the LR and batch size, further solidifying the argument for the proposed leverage scaling law.
- While the optimal configuration is quite impossible due to many hyperparameters, this paper still provides valuable and principled insights into how to scale the MoE.

**Weaknesses:**

- The experimental design for analyzing the Activation Ratio (A) (Section F.1, Table 8) appears structurally limited as it holds the active computational cost nearly constant.
  - The study varies the Activation Ratio (sparsity) exclusively by changing the total number of routable experts (E) while fixing the number of activated experts (Ea=2) and shared experts (Es=1). This approach means the study explores *only* the effect of increasing total parameters without changing the active expert.
- Formulating the scaling law in terms of the efficiency leverage is not very useful. This means that a user would have to first compute the dense model scaling law, before applying it to the proposed scaling law to figure out the MoE scaling behavior. A direct formula relating only the MoE configuration to the loss (eg [2501.12370]) would be more useful.
- Codes are not available (or a description of it if possible).
  - Training configuration is lacking as well (e.g., what kind of parallelism techniques is used etc)

**Questions:**

- How many GPU hours are used in the study?
- What kind of software (and techniques, like parallelisms) is used for training?
- What could be the reason for the difference wrt Olmoe in terms of the effectiveness of shared expert? In the Olmoe paper ([2409.02060]), it is shown that using shared expert is not effective.
- Using different number of head/ kv heads affects the number of model parameters. How does changing the kv head affect the results?
  - In table 8 it seems like the number kv head changes with size. What is the justification for it?
- Since both model size (N) and dataset size (D) are varied in the experiments, I wonder if the scaling law could be described in terms of N and D, which could be more expressive than using only compute (which is roughly 6*N*D), i.e., something similar to the chinchila-like scaling law?

**Details Of Ethics Concerns:**

I have an anonymity concern regarding the model names used in Figure 2, which presents a potential breach of the double-blind review policy.

The fitted lines are labeled as "(Ling Dense)" and "(Ling MoE)".

The main text mentions that the pre-training data is sourced from a large-scale multilingual corpus created by the Ling Team.

This explicit naming association linking the results to the "Ling Team" and the "Ling MoE/Dense" models could violate the double-blind submission guidelines.

---

> ### Author Response · Authors · 2025-11-22
> **Response to Weaknesses**
>
> > **Weakness 1: The experimental design for analyzing the Activation Ratio (A) appears structurally limited.**
>
> **Response:**  Thank you for the observation. Holding the active computational cost constant is **a deliberate design choice** central to our analysis of MoE efficiency. **Our goal was to isolate the effect of model capacity (total experts $E$) from computational cost (activated experts $E_a$).**
>
> By fixing $E_a$, we can demonstrate that increasing the number of available experts—and thus lowering the Activation Ratio—directly improves performance under a constant computational budget (FLOPs). This controlled experimental setup is crucial and conventional [1,2,3,4], because MoE architectures present a vast design space with numerous interacting hyperparameters (e.g., activation ratio, expert granularity). **Isolating variables in this manner makes a systematic analysis feasible; otherwise, the complexity and cost of the study would be unaffordable.**
>
> We will clarify this motivation in the revised manuscript.
>
> &nbsp;
>
> > **Weakness 2: Formulating the scaling law in terms of the efficiency leverage is not very useful.**
>
> **Response:** We respectfully disagree with the assessment of our work's usefulness. While a direct formula for loss is indeed a valid approach, we argue that formulating the scaling law in terms of **Efficiency Leverage (EL) offers a more direct and practical way to understand MoE behavior, which is the core of our study**. Our formulation is deliberately designed to answer a more fundamental question for practitioners: "How much more efficient is a given MoE model compared to its dense counterpart?" The key advantages of this approach are:
>
> 1. **EL is More Direct and Interpretable Metric**: EL directly quantifies the efficiency advantage of an MoE architecture. For instance, an EL of 2.0 at compute budget $C$ has a clear, intuitive meaning: the specific MoE architecture is twice as efficient as its dense counterpart at compute budget $C$. In contrast, an absolute loss value (e.g., 1.5) is dataset-specific, and its practical significance is often obscure (e.g., is a 0.1 loss reduction meaningful?). Our EL-based approach provides an architectural insight that is decoupled from the specific dataset.
>
> 2. **EL can Simplify Model Selection**: Comparing two different MoE architectures using traditional loss-based scaling laws would require fitting two separate, complex models, as the reviewer noted. Our approach streamlines this process. A user can simply use our formula (i.e., Eq. 5) or look up a table (e.g., Figure 1(b)) to directly compare the EL of different configurations, making it far more user-friendly for making architectural design choices.
>
> In short, while traditional laws predict **"what" the loss will be**, our formulation explains **"how much better" an MoE architecture is**. We believe this provides a more actionable insight and will clarify these benefits in the paper.
>
> &nbsp;
>
> > **Weakness 3: Codes are not available and training configuration is lacking.**
>
> **Response:** We thank the reviewer for pointing out these training details.
> 1. **We will release the key code essential for reproducibility upon acceptance**, including model architecture implementations and the main training pipeline. A release of the entire internal codebase is impractical due to its scale and the lengthy review process required for commercial constraints. However, the provided code will be  sufficient to validate all results presented in this paper.
> 2. **Our implementation is built on Megatron-LM and employs a hybrid parallel strategy** combining Expert Parallelism (EP), Tensor Parallelism (TP), and Pipeline Parallelism (PP) based on model size. For general training configurations, we have already listed them in **Appendix D of the initial manuscript**. The parallelism configurations for our representative models are as follows (detail architecture of model can be found in Table 8-10).
>
>     | model | $n_{layers}$ | $d_{model}$ | $d_{ffn}$ | $d_{expert}$ | $E$ | $E_a$ | EP | TP | PP |
>     |:---|---:|---:|---:|---:|---:|---:|---:|---:|---:|
>     | Experimental Model Example 1 | 8 | 384 | - | 96 | 256 | 8 | 8 | 1 | 1 |
>     | Experimental Model Example 2 | 8 | 512 | - | 128 | 256 | 8 | 8 | 1 | 1 |
>     | Experimental Model Example 3 | 10 | 640 | - | 160 | 256 | 8 | 8 | 1 | 1 |
>     | Experimental Model Example 4 | 14 | 768 | - | 192 | 256 | 8 | 8 | 1 | 1 |
>     | Experimental Model Example 5 | 16 | 1024 | - | 256 | 256 | 8 | 8 | 1 | 1 |
>     | Experimental Model Example 6 | 22 | 1280 | - | 320 | 256 | 8 | 8 | 1 | 2 |
>     | Dense-6.1B | 28 | 4096 | 14336 | - | - | - | - | 2 | 1 |
>     | MoE-mini (A0.8B) | 20 | 2048 | 5120 | 384 | 384 | 12 | 8 | 2 | 1 |
>
> We will clarify these configurations in the revised manuscript.

---

> ### Author Response · Authors · 2025-11-22
> **Response to Questions**
>
> > **Question 1: How many GPU hours are used in the study?**
>
> **Response:** Our study utilized a total of **~680k equivalent H800 GPU-hours**:
>
> *   **~360k hours for preliminary experiments**, including hyperparameter tuning and model-data allocation.
> *   **~200k hours for the main architectural scaling experiments**, which form the core of our contribution.
> *   **~120k hours for final validation runs**, such as the full training of our 16B MoE and dense models on 1T tokens.
>
> This represents a computational cost of over **1 million dollars** (assuming at \$2/H800_GPU-hour). We believe this large-scale investigation will provide the community with robust and valuable insights. We will add these details to the revised manuscript.
>
> &nbsp;
>
> > **Question 2: What kind of software (and techniques, like parallelisms) is used?**
>
> **Response:** Our training utilizes the **Megatron-LM with a hybrid parallel strategy**. The specific parallel configurations are listed in our response to Weakness 3.
>
> &nbsp;
>
> > **Question 3: What could be the reason for the difference wrt Olmoe in terms of the effectiveness of shared expert?**
>
> **Response:** We attribute the differing conclusions on shared expert  primarily to two fundamental differences:
>
> 1.  **Scope of Analysis (Scaling Laws vs. Single Data Point)**: OLMoE's findings are based on a specific model. In contrast, **our conclusion is derived from a broader scaling law analysis across various model sizes and compute budgets**. Intriguingly, **our own results align with OLMoE's for specific configurations**; for instance, as shown in Figure 3(c), the second-largest MoE configuration (M=2e9) achieves the lowest loss (2.183) without a shared expert (Ratio = 0%). However, the overall trend across scales favors using one shared expert as the generally optimal strategy.
>
> 2.  **Architectural Differences**: Our experiments on shared expert are based on a 256-expert MoE architecture, which features a **significantly higher sparsity and finer granularity** than OLMoE. This fundamental architectural difference, combined with variations in training hyperparameters and routing strategies, may substantially influence the final outcome.
>
> In short, the benefit of a shared expert appears context-dependent and our goal was to identify a more general scaling trend. We will add a discussion to the paper to contrast our findings with OLMoE's.
>
> &nbsp;
>
> > **Question 4: How does changing the number of head affect results, and why do KV heads change with model size?**
>
> **Response:** Changing the number of heads according to the variation of model size is a common practice[1,2]. Our additional experiments show that the **optimal number of heads scales with model size and typically lies within a range.** To ensure a fair comparison where each model operates near its peak performance, we vary the number of heads with model size to maintain the per-head dimension (e.g., 64) constant. The following table presents the near-optimal number of heads, which are defined as models achieving a validation loss within 0.5% of the minimum observed loss.
>
> | model | $n_{layers}$ | $d_{model}$ | $n_{head}$ | near-optimal number of heads |
> |:---|---:|---:|---:| ---:|
> | experiment_model | 8 | 512 | 8 | {4,8} |
> | experiment_model | 10 | 640 | 10 | {4,8,10} |
> | experiment_model | 12 | 768 | 12 | {8,12} |
> | experiment_model | 16 | 1024 | 16 | {8,16} |
>
> Due to limits on the OpenReview platform, detailed figures will be provided in the revised manuscript.
>
>
> &nbsp;
>
> > **Question 5: If the scaling law could be described in terms of N and D?**
>
> **Response:** Thank you for this insightful suggestion. We agree that a Chinchila-like scaling law would be more comprehensive. However, the vast design space of MoE models (e.g., sparsity, expert granularity, shared experts) makes jointly varying N and D to fit such a law **computationally infeasible** [5]: achieving **reliable** predictions would require a very large experimental grid across design points.
>
> Thus, we adopted total compute as the primary scaling variable. This provides a direct and tractable method for comparing MoE and dense models, **balancing experimental cost with generalizable insights**. We will add this as a key direction for future work, positioning our study as a foundational step.
>
> &nbsp;
>
> ---
>
> **References:**
>
> [1] Hoffmann, Jordan, et al. "Training compute-optimal large language models." arXiv preprint arXiv:2203.15556 (2022).
>
> [2] Krajewski, Jakub, et al. "Scaling laws for fine-grained mixture of experts." arXiv preprint arXiv:2402.07871 (2024).
>
> [3] Abnar, Samira, et al. "Parameters vs flops: Scaling laws for optimal sparsity for mixture-of-experts language models." arXiv preprint arXiv:2501.12370 (2025).
>
> [4] Clark, Aidan, et al. "Unified scaling laws for routed language models." International conference on machine learning. PMLR, 2022.
>
> [5] Besiroglu, Tamay, et al. "Chinchilla scaling: A replication attempt." arXiv preprint arXiv:2404.10102 (2024).

---

> ### Author Response · Authors · 2025-11-22
> **Response to Ethics Concerns**
>
> We thank the reviewer for this important feedback. The model name in question is **just a neutral, internal codename**. It carries no special meaning and **is not associated with any institution, product, or author identity**. To eliminate any potential ambiguity, we will replace all occurrences with a neutral placeholder in the latest version.

---

> ### Comment · Reviewer_k9rc · 2025-11-24
>
> Thank you for the response. I am largely satisfied with the clarification.
>
> Weakness 1: given the number of gpu hours already spent in the paper, the lack of investigation on changing topk is acceptable.
>
> Weakness 2: the use of loss ratio instead of the absolute value of the loss is indeed helpful in some scenarios.
>
> Question 4:
> You seem not to be answering the question about why kv head increases? If not mistaken, the no. of parameters is given in equation 10, and there are dependencies on both $n_h$ and $n_{kv}$.
> Your reply seems to be optimizing $n_h$ only and not $n_{kv}$.
> Sorry to be nit-picking; I think that it is completely fine that you are simply following the standard practice of increasing $n_{kv}$ without further experiments, given that a great amount of compute has already been spent. However, a clarification would be helpful.

---

> > ### Author Response · Authors · 2025-11-24
> >
> > Thank you for your quick reply. We are very pleased that we could largely address your concerns.
> >
> > &nbsp;
> >
> > Regarding Question 4, you are correct. A comprehensive analysis for $n_{kv}$ similar to our study on $n_{head}$ (testing ~4 values per size to map its scaling trend) would have been computationally prohibitive during the rebuttal, requiring an estimated 10,000 additional GPU-hours. Although $n_{kv}$ affects parameter count and compute, it was held fixed within each model size across all experimental groups, so we are confident this does not affect our conclusions about the MoE architecture. Therefore, for $n_{kv}$, we simply followed the standard practice of increasing its value with model size.
> >
> > We appreciate your rigorous feedback and will further clarify this point in the manuscript. We also plan to systematically investigate these design dimensions in our future work.
> >
> > &nbsp;
> >
> > Thank you again for your thoughtful review. We are committed to presenting the most solid and comprehensive results possible. If there is anything else that you would like us to clarify, please let us know!

---

### Author Response · Authors · 2025-11-24
**General Response and PDF Revision Summary**

We would like to thank all the reviewers for their valuable feedback. We are encouraged that reviewers found our core contribution, the Efficiency Leverage (EL) metric, to be an original and useful concept (Reviewer 7T91, WZ6e) that provides valuable and principled insights for scaling MoE models (Reviewer k9rc, 7T91). Reviewers also highlighted our systematic and principled methodology, which includes detailed ablation studies (Reviewer k9rc) and ensures fair comparisons by addressing methodological flaws in prior work (Reviewer 7T91). Finally, we are glad that the high empirical quality of our work was recognized, from its large scale (Reviewer k9rc) and extensive data (Reviewer 7T91) to the strong validation where our scaling law's predictions were successfully confirmed at a much larger scale (Reviewer 7T91, WZ6e).

&nbsp;

We would also like to highlight the significant computational investment this work represents, **``exceeding 680,000 equivalent H800 GPU hours``**, which corresponds to an estimated rental cost of **``over one million US dollars``**. It is our sincere hope that by undertaking such extensive experiments, we can provide the community with valuable and actionable insights into MoE architecture design.

&nbsp;

In response to the reviewers' insightful feedback, we have made several key revisions to enhance the clarity, rigor, and completeness of our paper. The main changes are summarized as follows:

* **Clarified the Definition and Formulation of EL (for Reviewer WZ6e):**

  We have refined the wording regarding the meaning of EL in Section 1 and supplemented this with a formal derivation and justification for its functional form in Section 3.

* **Strengthened Analysis of Expert Granularity (for Reviewer WZ6e):**

  We revised our claims about overly coarse and overly fine-grained experts in Section 4.1, supporting them with additional experimental results in Appendix F.2.

* **Improved Methodological Transparency (for Reviewer WZ6e):**

  We have detailed the methodology for obtaining EL data points in Section 4.2.1 and provided a more thorough explanation with pseudocode in Appendix K.

* **Added Goodness-of-Fit Analysis for Scaling Laws (for Reviewers WZ6e and 7T91):**

  In Section 4.2, we now present a comprehensive goodness-of-fit analysis, including R-squared, RMSE, and residual statistics. Further graphical analysis has been added to Appendix N.

* **Expanded Discussion on Related Work and Limitations (for Reviewers k9rc and 7T91):**

  We have added a comparison with OLMoE and further clarified the practical advantages of EL over traditional loss-based scaling laws in Section 6 and Appendix C (for Reviewers k9rc). In Section 7, we explicitly acknowledge the paper's limitations, such as MoE hyperparameter scaling laws and wall-clock time estimation  (for Reviewers 7T91).

* **Other Improvements (for Reviewers k9rc, WZ6e and 7T91):**

  We have also incorporated several other revisions, including retitling Section 4.2.1 to "Univariate Scaling Laws" (for Reviewer WZ6e), and adding details on our training techniques in Appendix D (for Reviewer k9rc), computational cost in Appendix L (for Reviewer k9rc), and an additional experiment on the number of attention heads in Appendix M (for Reviewer k9rc).

Once again, we thank all the reviewers for their time and insightful comments, which significantly improved our paper.

---

### Author Response · Authors · 2025-12-04
**Summary of Our Responses and Revisions during the Rebuttal Period**

Dear Area Chairs,

We sincerely thank you for your time and effort in handling our paper.

We conduct an **industrial-scale empirical study** on the scaling laws of MoE models, backed by a substantial investment of **`over 680,000 H800 GPU-hours`** (estimated to cost more than **`one million US dollars`** in rental fees). In this work, we introduce "Efficiency Leverage" (EL), a  concept to **quantify and predict the computational advantages of MoE** over dense models. These findings provide crucial support for the pre-training of **`trillion-parameter MoE models`**.

---

# 1. Reviewer Feedback
During the rebuttal period, our detailed responses and revisions were acknowledged by the reviewers:

* **`Review k9rc (6)`** praised the "large scale" of the study and the "principled approach" to tuning. After reviewing our detailed responses, they acknowledged that we had **largely addressed their concerns** and agreed that the EL metric is "indeed helpful."
* **`Review 7T91 (6, no further discussion)`** highlighted the originality of the "Efficiency Leverage" formulation and the high empirical quality supported by significant computational investment. We provided detailed justifications and additional data for their questions about fitting process and practical wall-clock costs.
* **`Review WZ6e (4, no further discussion)`** acknowledged the comprehensive three-phase methodology and raised major concerns regarding the definition of our core metric. We provided comprehensive clarifications, a formal derivation, and a detailed explanation of our experimental procedure. We regret that we were unable to receive their feedback on these extensive revisions.

---

# 2. Our Strengths
We are encouraged that the reviewers recognized the core contributions of our work:

* **Large-Scale and Rigorous Experiments:**  All reviewers acknowledged the scale of our empirical study, which consumed **~680k H800 GPU-hours to train over 300 models up to 28B parameters** and validated the accuracy of our scaling laws on a 17B MoE model trained with 1T tokens.

* **Novelty and Conceptual Contribution:** The novelty of our core metric, Efficiency Leverage (EL), was highlighted by Reviewers `7T91` and `WZ6e`, who recognized it as **a clear and useful tool for comparing MoE architectures**.
* **Systematic and Principled Methodology:** Reviewers `k9rc` and `7T91` specifically noted that our rigorous experimental setup, which establishes optimal hyperparameters and data allocation, **addresses a methodological flaw in prior work** and ensures fair comparisons.
* **Practical Guidance:** The work was praised for offering clear guidance despite the complexity of MoE training. Reviewers described it as providing "valuable and principled insights" (`k9rc`) and serving as a "**practical guide for designing future efficient MoE models**" (`7T91`).

---

# 3. Main Responses and Revisions
Below we summarize our responses and the corresponding revisions to the manuscript:

* **Strengthened Theoretical Foundations:**
  * **Clarified the EL Formulation (for Reviewer `WZ6e`):** We refined the definition of EL in Section 1 and added a formal derivation and justification for its functional form in Section 3.

  * **Added Goodness-of-Fit Analysis (for Reviewers `WZ6e` and `7T91`):** We added a comprehensive goodness-of-fit analysis for our scaling laws in Section 4.2, with further graphical analysis in Appendix N.


* **Additional Experiments and Analysis:**
  * **Expert Granularity (for Reviewer `WZ6e`):** We revised our claims about fine-grained experts in Section 4.1, supporting them with additional experiments in Appendix F.2.
  * **Attention Heads (for Reviewer `k9rc`):** We added an experiment in Appendix M to clarify how scaling the number of attention heads with model parameter counts.

* **Methodological Transparency:**

  * **Detailed Procedure for Obtaining EL (for Reviewer `WZ6e`):** We detailed the methodology for obtaining EL data points in Section 4.2.1 and provided a more thorough explanation with pseudocode in Appendix K.

  * **Training Techniques (for Reviewer `k9rc`):** we adding details on training techniques in Appendix D, such as the parallelism techniques and parallelism configurations.

* **Expanded Discussion and Other Improvements:**
  * **Related Work and Limitations (for Reviewers `k9rc` and `7T91`):** We added a comparison with OLMoE (Section 6, Appendix C) and explicitly discussed limitations, such as wall-clock time estimation and hyperparameter scaling (Section 7).
  * **Other Improvements (for Reviewers `k9rc` and `WZ6e`):** We reported computational costs in Appendix L and retitled Section 4.2.1 to "Univariate Scaling Laws" for clarity.

---

We believe these revisions substantially improve the rigor, clarity, and completeness of our work. **All changes are included in the revised manuscript and clearly highlighted.**


Thank you once again for your valuable time and consideration.

Best Regards,

Authors

---

### Meta-Review · Area_Chair_XXug · 2026-01-06

**Summary:**

### Strengths
* This paper is well structured.
* This work defines "efficiency leverage" metric to quantify MoE computational advantage.
* This experiments are systematic, structured, and of large scale.
* This work establishes empirical scaling law w.r.t. crucial MoE architecture parameters.
* This work also derives empirical scaling law for optimal learning rate and batch size.
* These empirical scaling laws provide valuable insights into how to scale MoE.
* The final validation, MoE-mini, is a solid verification of the insights.

### Concerns
1. Practicalness
	1. Formulating the scaling law in terms of efficiency leverage might not be useful as directly relating loss to the architectural parameters [Reviewer k9rc].
	2. The conclusion on efficiency monotonically increases with sparsity is based on #FLOPs instead of practical wall-clock cost of routing, communication and memory bandwidth for loading distinct expert weights, which can become bottlenecks at high sparsity [Reviewer 7T91].
2. Soundness
	1. The functional form for the scaling law has a complex form and lacks strong justification [Reviewer 7T91].
	2. Imprecise and inconsistent definition of EL [Reviewer WZ6e].
	3. Technical issues on some formula and symbol [Reviewer WZ6e].
	4. The claim about expert granularity is stated as fact but lacks supporting analysis or evidence in the paper [Reviewer WZ6e].
3. Clarity and resource
	1. Details of obtaining datapoints for EL should be clarified [Reviewer WZ6e].
	2. Unclarity of Figure 4 [Reviewer WZ6e].
	3. Code and training settings are not available [Reviewer k9rc].

**Reviewer Concerns:**

* The concerns on practicalness are addressed or answered well.
* For concerns on soundness, the authors provide detailed answers to the reviewers, clarifying the fitting process, imprecise definition, and unsupported claims.
* Concerns on clarity and resource are also answered well.

**Reviewer Scores:**

* Reviewer WZ6e initially gave 4. I guess there is an ~60% chance that the reviewer will increase the score.
* Reviewer k9rc and 7T91 initially gave 6. I think they will at least keep the positive score.

---

### Decision · Program_Chairs · 2026-01-26

Accept (Poster)